# Correlating fluorescence microscopy, optical and magnetic tweezers to study single chiral biopolymers such as DNA

Jack W. Shepherd[1,2,6], Sebastien Guilbaud[1,6], Zhaokun Zhou[3,6], Jamieson A. L. Howard[1,6], Matthew Burman[1], Charley Schaefer[1], Adam Kerrigan[4], Clare Steele-King[5], Agnes Noy[1] & Mark C. Leake[1,2] ✉

Biopolymer topology is critical for determining interactions inside cell environments, exemplified by DNA where its response to mechanical perturbation is as important as biochemical properties to its cellular roles. The dynamic structures of chiral biopolymers exhibit complex dependence with extension and torsion, however the physical mechanisms underpinning the emergence of structural motifs upon physiological twisting and stretching are poorly understood due to technological limitations in correlating force, torque and spatial localization information. We present COMBI-Tweez (Combined Optical and Magnetic BIomolecule TWEEZers), a transformative tool that overcomes these challenges by integrating optical trapping, time-resolved electromagnetic tweezers, and fluorescence microscopy, demonstrated on single DNA molecules, that can controllably form and visualise higher order structural motifs including plectonemes. This technology combined with cutting-edge MD simulations provides quantitative insight into complex dynamic structures relevant to DNA cellular processes and can be adapted to study a range of filamentous biopolymers.

Multiple cell processes involve chiral biopolymers experiencing pN scale forces[1] and torques of tens of pN.nm[2], exemplified by molecular machines on DNA[3] where torsion is a critical physical factor[4]. Although lacking torsional information, optical tweezers (OT)[5] combined with fluorescence microscopy of dye-labelled DNA was used to image DNA extension in response to fluid drag[6], and though missing topological details from fluorescence visualisation, magnetic tweezers (MT) experiments have demonstrated how DNA molecules respond to force combined with supercoiling[7]. Developments in OT and MT have enabled molecular study of DNA over-stretching[8], protein binding[9], dependence of mechanical properties to ionicity[10], and DNA-protein bridge formation[11]. Fluorescence microscopy combined with OT have enabled super-resolved measurement of extended DNA in the absence

of torsional constraints[12,13], with angular OT enabling torque control[14]. High-precision MT has facilitated single-molecule DNA study of over- and undertwisting, including twist-stretch coupling[15], dependence of torsion on temperature[16] and salt[17], and binding of filament-forming proteins such as RecA[18]. MT has enabled single-molecule torsion dependence measurements of non-canonical DNA structures including P-DNA[19], left-handed DNA[20], and higher-order motifs of the G-quadruplex[21] and plectonemes[22], applying vertical geometries to extend molecules orthogonal to the focal plane. DNA plectonemes have been visualised using fluorescence microscopy combined with MT, by reorienting molecules almost parallel to the focal plane, limited to observations *following* but not *during* plectoneme formation[20]. Plectoneme formation has also been induced through intercalator

[1]School of Physics, Engineering and Technology, University of York, York YO10 5DD, England. [2]Department of Biology, University of York, York YO10 5DD, England. [3]Guangdong Provincial Key Lab of Robotics and Intelligent System, Shenzhen Institute of Advanced Technology, Chinese Academy of Sciences, Shenzhen, China. [4]The York-JEOL Nanocentre, University of York, York YO10 5BR, England. [5]Bioscience Technology Facility, University of York, York YO10 5DD, England. [6]These authors contributed equally: Jack W. Shepherd, Sebastien Guilbaud, Zhaokun Zhou, Jamieson A. L. Howard. ✉e-mail: mark.leake@york.ac.uk

binding to DNA combed onto surfaces then imaged with fluorescence microscopy to track plectonemes[23].

OT combined with fluorescence have been used to study DNA over-stretching in torsionally unconstrained DNA[24], revealing S-DNA structures by measuring polarised emissions from bound fluorophores[25]. Later developments using torsional constraints were implemented by stochastic stretch-unbind-rebinding applied to single optically trapped DNA molecules, with a caveat that supercoiling cannot be controlled in advance, or reversed[26].

Simulations have enabled insight into DNA structural transitions, coarse-grained approaches using oxDNA[27] on mechanically[28–30] and thermally perturbed DNA[15], and all-atom methods which predict sequence-dependence to torsionally-constrained stretching[31] and explore the denaturing pathways of torsionally unconstrained stretching[32]. Simulations have also been used with single-molecule MT via space-filling algorithms to predict P-DNA formation[19] and molecular dynamics (MD) simulations to study stretch/twist coupling[15]. More recently, DNA minicircles comprising just a few hundred base pairs (bp) have been used as computationally tractable systems to investigate supercoiling[33] and using AFM imaging[34].

Single-molecule experiments used to probe DNA mechanical dependence on structural conformations have advantages and limitations. OT enables high forces beyond the ~65 pN over-stretching threshold up to ~1 nN[35]. However, they cannot easily control torque of extended molecules without non-trivial engineering of trapping beams and/or the trapped particle's shape[36], and establishing stable torque comparable to MT using readily available microbeads is not feasible. While MT can enable reversible supercoiling of single molecules at ~pN forces[7,19], mechanical vibrations over ~seconds introduced during rotation of nearby permanent magnets places limitations on structural transitions probed.

We present Combined Optical and Magnetic Biomolecule TWEE-Zers (COMBI-Tweez), which overcomes these challenges. COMBI-Tweez is a bespoke, correlative single-molecule force and torsion transduction technology colocating low stiffness near-infrared (NIR) OT with laser excitation fluorescence microscopy on DNA molecules in a ~mT B field, generated by pairs of Helmholtz coils, rotated in orthogonal planes independently under high-precision control (Supplementary Movie 1). OT and MT can be operated independently, while a trapped fluorescently-labelled DNA molecule, which we use as well-characterised test chiral biopolymer, can be extended parallel to the focal plane to enable fluorescence imaging in real-time with sub-nm displacement detection via laser interferometry. To our knowledge, this is the first report of stable optical trapping of magnetic beads at physiologically relevant temperatures.

We demonstrate this technology through time-resolved formation and relaxation of higher-order structural motifs including plectonemes in DNA; these emergent features comprise several open biological questions which relate to plectoneme size, position and mobility. We measure changes to the buckling transition[37] due to binding of the fluorescent dye SYBR Gold[38], and controlled quantification of interactions between two "braided" DNA molecules. We discuss these findings in the context of modelling structural motifs using MD simulations. The capability of COMBI-Tweez is the correlative application of several single-molecule techniques on the same biopolymer molecule, not attainable with a subset of techniques alone. OT enables high-bandwidth force measurement, while MT generates magnetic microbead rotation to induce controlled torque with no mechanical noise or lateral force. Control software enables precise tuning of supercoiling density $\sigma$, with bead rotation verified using fluorescence imaging of conjugated reporter nanobeads. We demonstrate COMBI-Tweez using high-sensitivity fluorescence detection whose utility can be easily extended to multiple fluorescence excitation and surface functionalization methods, with future applications to study several generalised biopolymers.

## Results

### Decoupling force and torque

COMBI-Tweez OT control of force and displacement, with MT control of torque (Fig. 1a, b, Supplementary Movie 1) is simple and robust. No modifications to NIR laser trapping beams are required, such as Laguerre-Gaussian profiles to impart angular momentum[39], nor nanoscale engineering such as cylindrical birefringent particles[40]. Since COMBI-Tweez OT controls bead displacement, dynamic B field gradients that update continuously to reposition the bead, common in MT-only systems[41], are not required. We use a uniform B field via two pairs of coils carrying sinusoidal currents to rotate an optically trapped magnetic bead (Fig. 1c, "Methods" section, Supplementary Note 1 and Supplementary Figs. 1–8). Decoupling force and torque allows COMBI-Tweez to exert arbitrary combinations of sub-pN to approximately 10 of pN and of sub-pN·nm to tens of pN·nm, relevant for cellular processes involving DNA such as plectoneme formation[42]. COMBI-Tweez also exploits rapid 50 kHz quantification of bead positions using back focal plane (bfp) detection of the NIR beam via a high-bandwidth quadrant photodiode (QPD) not limited by camera shot noise[43] ("Methods" section). In MT-only setups, camera-based detection is often used to track beads with ~kHz sampling; our high-speed detection permits faster sampling to probe rapid structural dynamics.

### Stretching/twisting label-free DNA

We first assessed COMBI-Tweez to perform torsionally unconstrained stretch-release (MT module off) using a 15.6 kbp test double-stranded DNA (dsDNA) construct in the absence of fluorescent tags ("Methods" section, Supplementary Fig. 9), generated as a fragment from λ DNA digestion and functionalised with concatemer repeats of biotin or digoxigenin "handles" on either end, since the results can be directly compared with several previous OT/MT molecular studies on similar constructs. Our handles were 500 bp leaving a 14.6 kbp central region. We surface-immobilised 5 μm "anchor" beads with non-specific electrostatic adsorption by flowing diluted anti-digoxigenin (DIG) beads into a flow cell before BSA passivation then introducing DNA in ligase buffer. Following incubation and washing we introduced 3 μm Neutravidin-functionalised magnetic beads, sealed the flow cell and imaged immediately. To form tethers, we hold an optically trapped magnetic bead 500 nm from an anchor bead to allow a tether to form through in situ incubation, which we confirmed in preliminary experiments across a range of buffering conditions and ionic strengths from approximately 10–200 mM NaCl, prior to focusing just on physiological conditions using phosphate buffer saline ("Methods" section). Each tether's mechanical properties could then be studied via stretch-release through nanostage displacement at a ~1 Hz frequency and ~5 μm amplitude ("Methods" section and Supplementary Movie 2).

Since it is well established that strong B fields induce above physiological temperature increases in magnetic nanoparticles and also that previous attempts using optically trapped magnetic beads were limited by temperature increases due NIR laser absorption by magnetite[44] (and personal communication Nynke Dekker, University of Oxford), we characterised trapped bead temperature extensively. Performing transmission electron microscopy (TEM) on dehydrated resin-embedded 70 nm thick sections of the magnetic beads ("Methods" section) indicated electron opaque nanoparticles distributed in a~100 nm thin surface just inside the outer functionalisation layer (Fig. 2a, b) while electron diffraction was consistent with an iron (II/III) oxide magnetite mixture as expected (Fig. 2c). We estimated temperature increase near trapped beads by measuring the distance dependence on melting alkane waxes ("Methods" section). By positioning a trapped bead adjacent to a surface-immobilised wax particle, we observed the melting interface using brightfield microscopy (Fig. 2d–f), enabling distance measurement to the bead surface (Fig. 2g). Modelling heat generation analytically suggested ~$1/x$ dependence for small $x$ (distance from the bead surface) less than the

bead radius (Supplementary Note 2). We used the data corresponding to small $x$ to show that increasing the distance of the magnetite from the bead centre significantly reduces the bead surface temperature (Supplementary Fig. 10), resulting in 45 °C in our case (Supplementary Note 3). Finite element modelling of heat transfer outside the bead confirmed deviation of $1/x$ for larger $x$ (Fig. 2h), fitting all wax data points within experimental error, indicating a range of 45–30 °C for distances from the bead surface from zero through to the ~5 μm DNA contour length. The mean ~37–38 °C over this range is serendipitously perfectly aligned for physiological studies. This acceptable temperature increase limits our trapping force to approximately 10 pN, well within the physiological range we want to explore for DNA.

The estimated DNA persistence length determined from worm-like chain (WLC)[45] fits to force-extension data ("Methods" section) was comparable to previous studies[46] with mean of approximately 50 nm (Fig. 3a) with no significant difference between stretch and release half cycles indicating minimal hysteresis due to stress-relaxation[47] (Supplementary Fig. 13a). Mean contour length was approximately 5 μm (Fig. 2a), comparable to sequence expectations. We configured OT stiffness to be low (10 pN/μm) compared to earlier DNA studies that probed overstretch, allowing instead stable trapping up to a few pN relevant to physiological processes while avoiding detrimental heating.

We tested COMBI-Tweez to twist torsionally-constrained DNA, first keeping the OT stationary as a trapped bead was continuously rotated at 1 Hz. Upon prolonged undertwist at a starting force of 1–2 pN the subsequent force remains broadly constant consistent with melting of the two DNA strands, whereas performing the same experiment using prolonged overtwist we found that the force increased as DNA is wound, until high enough to pull the bead from the

trap, towards the anchor bead at which point meaningful measurement ends (Supplementary Movie 3). As reported by others, we observed that DNA force does not increase linearly with torsional stress upon overtwist but undergoes non-linear buckling[37] at which DNA no longer absorbs torsional stress without forming secondary structures which shorten its end-to-end length (Fig. 3b). Utilising the 50 kHz QPD sampling, we monitored rapid fluctuations in conformations between individual rotation cycles which would not be detectable by the slower video rate sampling (Fig. 3b inset), with a-2-fold increase in the fluctuation amplitude during buckling which may indicate rapid transitioning of ~100 nm metastable regions of DNA. For the example shown, there is an initial 1.7 pN force at 4.5 μm extension (bead image inset, 36 s). Force increased non-linearly during rotation with corresponding decrease in extension to 4.3 μm (inset, 72 s) prior to buckling at 2.2 pN and exiting the trap (75 s). Performing power spectral analysis indicated an 85% increase in low frequency (<60 Hz) components during buckling, qualitatively consistent with the emergence of cumulatively supercoiled DNA with higher overall frictional drag. Modelling twist for an isotropic rod suggests torque τ per complete DNA twist is approximately $C/L$ where $C$ is the torsional modulus and $L$ the DNA length. Using $C = 410$ pN.nm obtained previously on DNA using MT single-molecule twist experiments[2] indicated torque of 0.08 pN.nm per bead rotation.

We next sought to reproduce "hat curves" which map relations between DNA extension versus twist when positively and negatively supercoiled ("Methods" section). At forces less than a critical value $F_c$ of 0.6–0.7 pN[48], DNA supercoiling was approximately symmetric around $σ = 0$[49]. At higher forces, negative supercoiling no longer leads to plectonemes but instead to melting bubbles[48]. We selected forces either side of $F_c$, fixed in real-time using bead position feedback to the

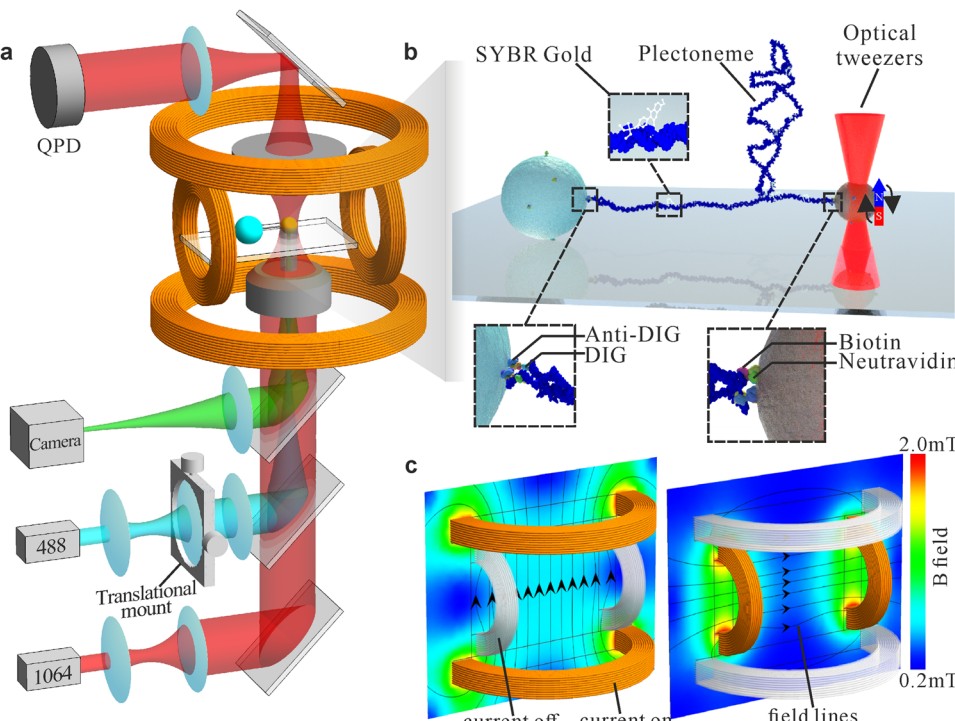

**Fig. 1 | Principle of COMBI-Tweez. a** An NIR OT is colocated with MT generated from a pair of orthogonal Helmholtz coils enabling a single tethered DNA molecule to be visualised simultaneously in the focal plane using a variety of light microscopy modes, to observe DNA structural dynamics when the molecule is mechanically perturbed. **b** Cartoon of a plectoneme formed in COMBI-Tweez, using chemical conjugation to tether one end of a DNA molecule to a surface-immobilised "anchor" bead with the other tethered to an optically trapped magnetic microbead. SYBR Gold fluorophores[38] label the DNA via intercalation between adjacent base pairs. **c** Simulation of B field when vertical and horizontal coil pairs are separately activated, indicating a highly uniform field in the region of a trapped bead. Generalised magnetic field vectors can be generated by combining outputs from each coil; it is simple to generate stable B field rotation in a plane perpendicular to the microscope focal plane resulting in rotation of the trapped bead that enables torque to be controllably applied to a tethered DNA molecule.

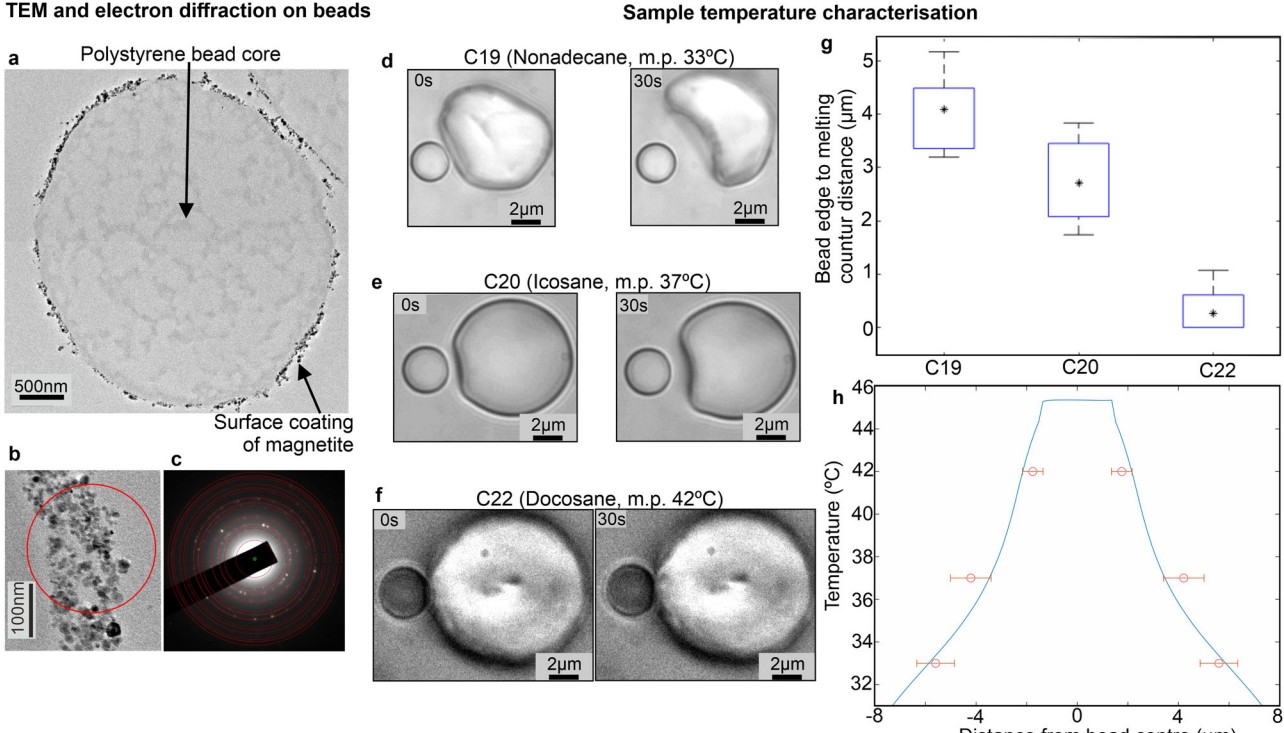

**Fig. 2 | Absorption of NIR laser by magnetite results in physiological temperatures surrounding trapped beads. a** TEM of 70 nm thick section taken from one resin-embedded dehydrated magnetic bead (JEOL 2100 + TEM 200 kV). **b** Zoom-in on the bead surface showing red circled electron-dense nanoparticles produce a **c** selected area diffraction pattern consistent with iron (II, III) oxide. Over 50 beads were viewed across three sectioned samples with identical form by eye. Select area diffraction was done in several locations, each showing an iron (II, III) oxide mix. **d–f** One representative of a brightfield of a trapped magnetic bead positioned adjacent to solid ~10 μm surface-immobilised alkane wax particle, shown for waxes C19, C20 and C22, indicating formation of melting interface at 30 s timepoint. C22 often either did not visibly melt or had interface less than a radius

from bead edge **g** Boxplot indicating mean (*) distance from bead edge to melting interface for the three waxes (blue box interquartile range, the whiskers indicate the data range, with the numbers of wax particles analysed $n_{C19} = 6$, $n_{C20} = 7$, $n_{C22} = 12$). **h** Finite element model (blue, Supplementary Note 3) for heat transfer predicts that the temperature stays constant inside the bead at ~45 °C and decreases with distance away from the bead (experimental data points overlaid in red, mean and s.d. error bars shown, from same datasets as panel (**g**)). The temperature dependence beyond bead surface becomes shallower than the ~1/$x$ predicted from analytical modelling (Supplementary Note 2) due to heat transfer into the glass coverslip from the surrounding water.

nanostage. As anticipated, high-force hat curves (Fig. 3c left panel, 1.1 pN) indicate asymmetry with a high extension plateau from σ = 0 to σ < 0 (blue trace) and σ > 0 to σ = 0 (green trace), with decreasing extension as DNA is overtwisted (red). At lower force (Fig. 3c right panel and Fig. 3d, 0.5 pN) negative supercoiling, indicated by green and red traces, comprised 200 rotations for each trace equivalent to maximum negative torque of −1.6 pN.nM, was broadly symmetrical with the positive supercoiling pathway, indicated by blue and yellow traces, which comprised 200 rotations per trace equivalent to maximum positive torque of +1.6 pN.nM. Lateral (15 nm/s) and axial (7 nm/s) drift was effectively negligible for individual rotation cycles by allowing COMBI-Tweez to reach a stable temperature following coil activation, however, over longer duration hat curve experiments comprising several hundred seconds we performed drift correction on bead positions ("Methods" section).

### DNA dynamics during stretch/twist visualised by correlative fluorescence imaging

We tested the fluorescence capability using several illumination modes including widefield epifluorescence, Slimfield[50] (a narrowfield microscopy enabling millisecond imaging of single fluorescent molecules), and an oblique-angle variant of Slimfield that combines narrowfield with highly inclined and laminated optical sheet microscopy (HILO)[51], using 488 nm wavelength laser excitation ("Methods" section). To visualise tether formation, we pre-incubated DNA-coated anchor beads with intercalating dye SYBR Gold[38] at one molecule per 1–2 μm²

(Fig. 4a, "Methods" section and Supplementary Movie 5), enabling inter-bead tethers to be visualised, using oxygen scavengers to mitigate photodamage (Fig. 4b). Low power widefield epifluorescence allowed single tethers to be observed at video rate for ~100 s before tether breakage due to photodamage (Fig. 4c). Repeating the same torsional unconstrained stretch-release protocol as for unlabelled DNA (Fig. 4d and Supplementary Movie 6) indicated a mean persistence length comparable to unlabelled of approximately 50 nm, whereas the contour length increased over 30% (Fig. 4e), consistent with earlier reports at comparable stoichiometry values of SYBR Gold:DNA[38] if at least half available SYBR Gold intercalation sites are occupied.

Experimenting with DNA concentration also demonstrated potential for studying interactions between multiple molecules, e.g. "braided" double tethers that can be controllably unwound by bead rotation to reform separated single tethers (Fig. 4f, Supplementary Fig. 14, "Methods" section and Supplementary Movie 7), including studying two fully braided coaxial DNA molecules (Fig. 4g and Supplementary Movie 8) in some instances resulting in visible entropic retraction when one of these snaps (Fig. 4h and Supplementary Movie 9). These distinct features of COMBI-Tweez allowed high-precision control of DNA-DNA interactions as a function of supercoiling.

### Time-resolved visualisation of DNA structural motifs

To explore the capability to study complex structural dynamics, we investigated DNA plectonemes, higher-order structural motifs that

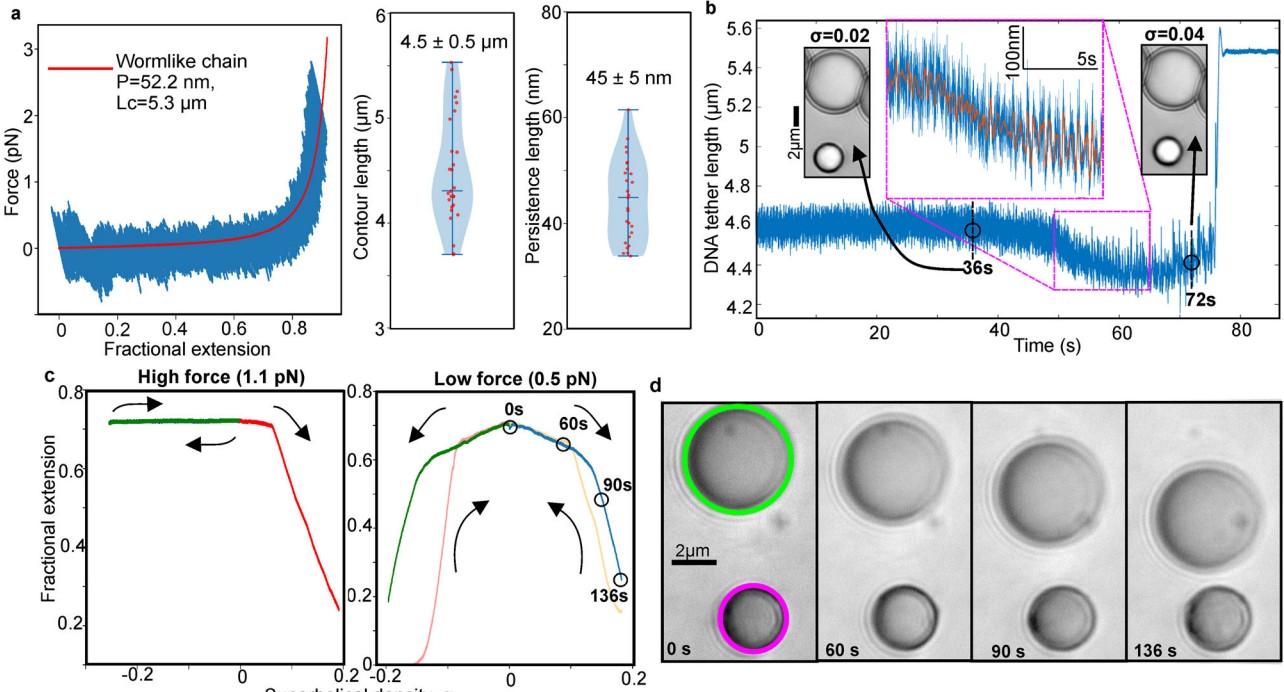

**Fig. 3 | COMBI-Tweez in brightfield quantifies key supercoiling features. a** Left: (Blue) representative DNA force vs extension showing no obvious stress-relaxation[47], WLC fit (red) Lc = 5.3 μm and persistence length $P$ = 52 nm obtained as mean from $n$ = 44 consecutive stretch-release half cycles, s.d. 13%, error on individual fits <0.002% estimated from 1000 bootstrapped fits from 1% of data per bootstrap. Right: violin plot of WLC fitted mean Lc and P for $n$ = 27 separate tethers, presented as mean value ± s.e.m. **b** Extension measured from QPD of trapped bead (blue) plotted as function of time during continuous overtwist for fixed trap, initially 1.7 pN. Inset: brightfield of bead pair at σ values of 0.02 and 0.04 taken at 36 s/72 s before magnetic bead lost from trap at 75 s. Zoom-in (blue) plus indicative running median overlaid (red, 5000-point window), showing same trend in decreasing extension as video analysis but revealing additional conformational heterogeneity between individual rotation cycles. Qualitatively similar non-linear buckling responses observed from $n$ = 5 overtwisted tethers pulled out of optical trap at a comparable force level to example shown here. **c** Left: Hat curve for force clamp experiment, 1.1 pN (green "outward" trace is almost entirely overlaid onto blue "inward" trace indicating negligible hysteresis); Right: hat curve, 0.5 pN. Both curves drift-corrected between different sections taken at time points which span up to 25 min. Apparent hysteresis between blue/yellow sections at low force due to 1–2 s response time of force clamp, while that between green/red sections is due both to this effect and bead pair transiently adhering for a few seconds following formation of tether extension reduction upon negative supercoiling. Arrows indicate path taken during the experiment (blue outward and yellow inward overtwist, followed by green outward and red inward undertwist). **d** Brightfield from low force clamp experiment of (**c**) at indicated time points, with nanostage moving anchor bead (green) relative to optically trapped bead (magenta) to maintain constant force (see also Supplementary Movie 4).

emerge in response to torque[46]. Applying constant force to a SYBR Gold labelled tether using the same protocol as for unlabelled DNA of 1–2 pN, above the critical $F_c$ of 0.6–0.7 pN[48], and rotation to control σ, resulted in a similar asymmetrical hat curve ("Methods" section) – in the example shown in Fig. 5a starting from σ = 0 and overtwisting (blue trace) to σ > 0 with associated drop in fractional extension, then undertwisting through to σ = 0 (yellow trace) then to σ < 0 with characteristic plateau in fractional extension (green/red traces), high-frequency QPD sampling enabling detection of rapid structural fluctuations over ~100 nm in this region (Fig. 5a inset). We measured a small difference to σ at which extension begins to decrease during continuous overtwisting; for SYBR Gold labelled DNA this occurred at σ close to zero, whereas for unlabelled DNA this was between 0.05 and 0.1, likely caused by DNA mechanical changes due to SYBR Gold intercalation[38].

To measure longer timescale structural dynamics, we redesigned data acquisition using stroboscopic imaging to expose just 10 consecutive images in fluorescence between 50 continuous bead rotations to minimise photodamage, allowing 1 min equilibration between 50 rotation segments – the example of Fig. 5b obtained below $F_c$ using this discontinuous imaging protocol showed the same qualitative features of an approximately symmetrical hat curve at low force for unlabelled DNA, with the caveat of negligible hysteresis following equilibration between rotation segments.

Using the stroboscopic illumination protocol, high forces for relaxed or negatively supercoiled DNA caused no differences in terms of tether shortening from the fluorescence images and no evidence of the formation of higher-order structures (Fig. 5c left and centre panels), though negatively supercoiled DNA exhibited image blur in localised tether regions, absent from relaxed DNA, manifest in a higher average full width at half maximum (FWHM) intensity line profile taken perpendicular to the tether axis (Fig. 5d and "Methods" section). Conversely, overtwist resulted in visible tether shortening along with the appearance of a higher intensity fluorescent puncta along the tether itself (Fig. 5c right panel and inset indicated as a plectoneme, see also Supplementary Movie 10. Significant overtwisting leads eventually to sufficient DNA shortening which pulls the magnetic bead out of the optical trap, shown in a different tether in Supplementary Movie 11).

At low force, fluorescence images of positively supercoiled DNA indicated similar tether shortening and formation of a fluorescent punctum (Fig. 5e and inset, σ = 0.11 and 0.14). Conversely, although relaxed DNA showed no evidence of higher-order localised structures, negatively supercoiled DNA indicated both tether shortening and the appearance of a fluorescent punctum (Fig. 5e and inset, σ = −0.14), albeit dimmer than puncta observed for positively supercoiled DNA. Notably, we find that tethers formed by undertwisting appeared at different locations from those created through positive supercoiling.

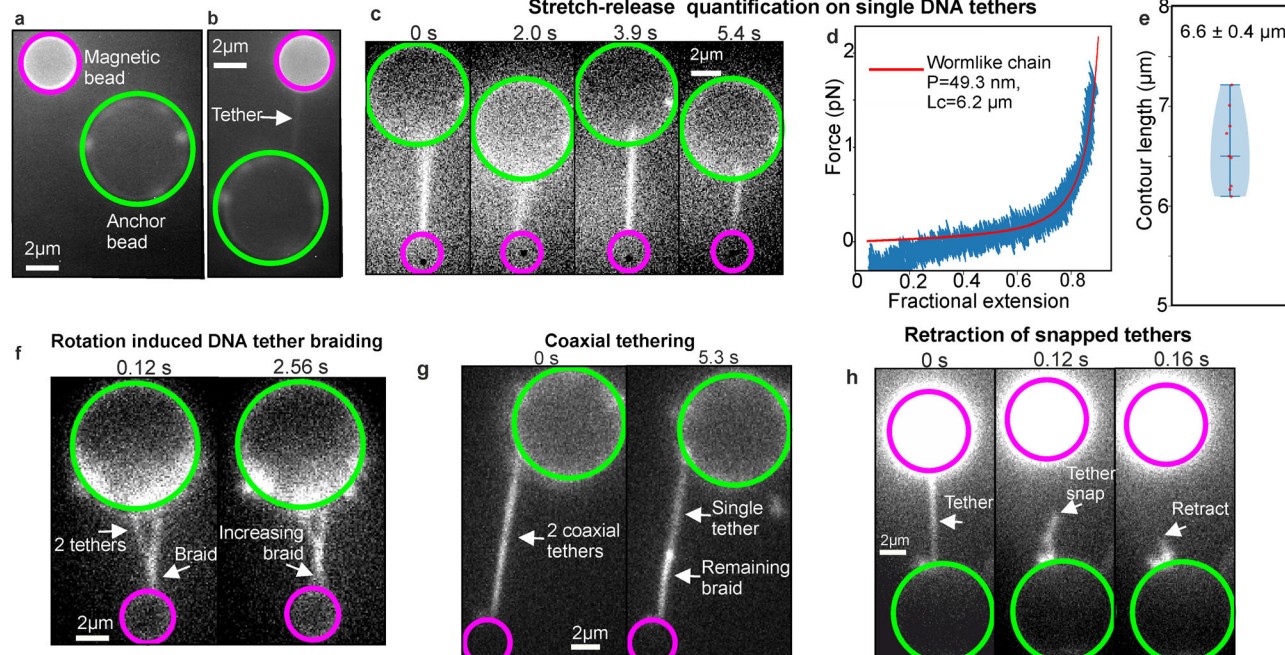

**Fig. 4 | Correlative fluorescence imaging and torsional manipulation of DNA enables visualisation of tether dynamics. a** One representative example of an optically trapped bead (magenta) next-to surface-immobilised anchor bead (green). Multiple DNA molecules clearly visible on anchor bead as distinct foci since entropic compaction of untethered DNA results in ~500 nm end-to-end length (also see Supplementary Fig. 11). **b** One representative example of a tether which is formed between the two beads in (**a**), indicated by white arrow. **c** One representative example of a tether showing image frames extracted from continuous stretch-release experiment performed in fluorescence. **d** Representative force-extension data compiled from $n = 8$ consecutive stretch-release half cycles obtained from same tether by WLC fits (red). Mean persistence length determined from $n = 9$ separate tethers was $53 \pm 15$ nm (±s.e.m.), comparable with that obtained

for unlabelled DNA construct while (**e**) mean contour length larger by >30% consistent with reports of SYBR Gold interactions[38] presented as the mean value ± s.e.m. from $n = 9$ independent tethers. **f** One representative example of two DNA molecules formed between anchor and trapped bead in which the tethers are coaxial. After one tether breaks (right), it retracts along the second leaving a bright still-braided region and a dimmer single-tether region. **g** One representative example of two DNA molecules which have formed a Y shape with "braided" section and two single tethers meeting anchor bead at different points. After rotation the braided section grows (see also Supplementary Fig. 14). **h** One representative example of a tether snapping and retracting to anchor bead, indicated by white arrow. All experiments were independently repeated at least 9 times.

Similar experiments on full length λ DNA (48.5 kbp) resulted in more than one plectoneme ("Methods" section), the example of Fig. 5f following positive bead rotations (Supplementary Movie 12) having three puncta per tether. Using Gaussian fitting to track puncta to 20 nm precision and summing background-corrected pixel intensities enabled estimation of bp content, indicating 2.6–3.1 kbp per plectoneme whose rate of diffusion decreased with increasing bp content (Fig. 5g). In this example, there was a step-like increase in DNA extension of ~400 nm at approximately 860 ms, consistent with a single-strand nick. Following this, each plectoneme then disappeared in sequence between approximately 900–1000 ms (Fig. 5h) indicative of a torsional relaxation wave diffusing from the nick[52].

Estimating the proportion of bp associated with puncta in the shorter 15 kbp construct for positively supercoiled DNA indicated up to ~50% of bp for the brightest puncta (σ = 0.14) and ~40% for the σ = 0.11 case are associated with plectonemes, while for negatively supercoiled DNA the brightest puncta contained closer to ~10% of the number of bp in the total tether length, broadly comparable with the range of bp per plectoneme in λ DNA. Puncta tracking indicated displacements of approximately 100 nm per frame equivalent to an apparent 2D Brownian diffusion coefficient in the focal plane of 310–1900 nm²/s (Table 1 and Fig. 6a). Note, that these 2D diffusion coefficient estimates give indications of the local mobility of plectonemes in the lateral focal plane of a DNA tether, though the physical processes of plectoneme mobility parallel (possible sliding movements) and perpendicular (likely to be lateral tether fluctuations in addition to plectoneme mobility relative to

tether) to a tether are likely to be different, which we do not explore here. We found that puncta for positively supercoiled DNA have higher mean diffusion coefficients than for negatively supercoiled by a factor of ~6.

We next predicted plectoneme location using a model based on localised DNA curvature[53] and the stress-induced destabilization (SIDD) during undertwist algorithm[54] to model the likelihood of forming melting bubbles and plectonemes at each different bp position along the DNA ("Methods" section), using the same sequence of the 15 kbp construct and so presenting an excellent opportunity for cross-validation between experiment and simulation. This analysis showed agreement for bubble formation to the location of puncta observed during negative supercoiling at low force off-centre at ~4.5 kbp (within 3.6% of our predictions, Fig. 6b). SIDD focuses on locating denaturation bubbles and has limitations in assuming torsional stress is partitioned into twist, therefore over-predicting bubble prevalence in systems in which plectonemes are present, and failing to account for influences of DNA curvature on plectonemes, and by extension, bubble formation. The presence of a predicted peak in the centre 7 kb region was likely an artefact of over-prediction, borne out by the integrated area under the suite of ~6 peaks pooled around ~4.5 kbp being over five times greater than the 7 kbp peak; it is likely this secondary bubble might have effectively been "replaced" by the plectoneme. Considering the high energy cost associated with initialisation of a run of strand separation of 10.84 kcal/mol[55], which is independent of bubble size, the formation of a single bubble is favourable in this regime.

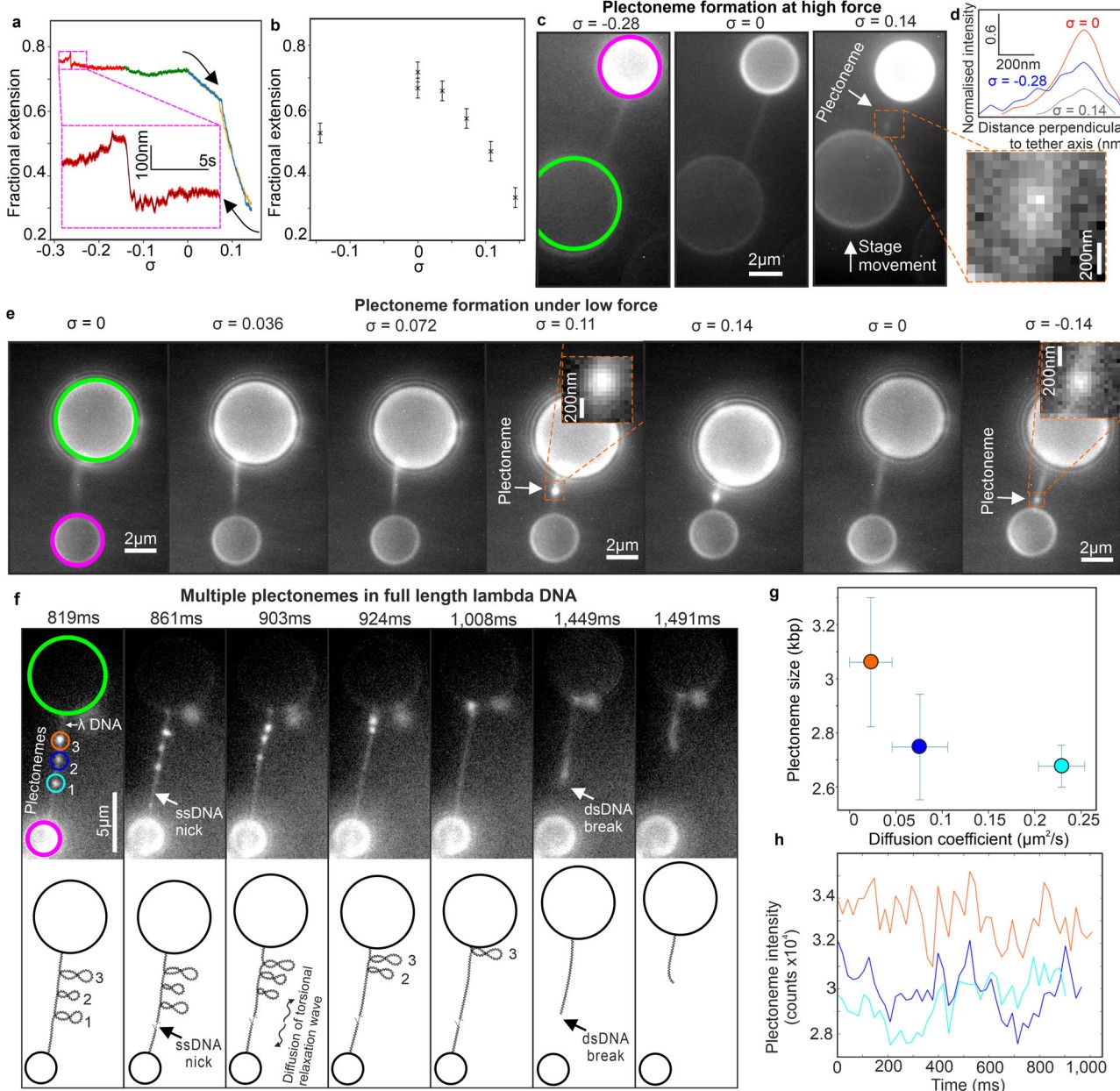

**Fig. 5 | Plectoneme dynamics in single molecules. a** 1.0 pN continuous asymmetric hat curve, SYBR Gold added (arrows indicate red-green-blue overtwist followed by yellow undertwist, drift-corrected). Inset: Zoom-in (red) plus black running 1500-point median filter. **b** 0.5 pN discontinuous hat curve showing extension reduction during positive/negative supercoiling (optical resolution error bars ± 0.03 fractional extension), n = 1. **c** Fluorescence microscopy of 1 pN plectoneme formation at σ = −0.28 (no plectoneme), 0 (no plectoneme) and 0.14 (inset plectoneme), one representative example. **d** Line profiles for tethers of (**c**) normalised to σ = 0 spanning blur region of σ = −0.28 tether using same size region for σ = 0 and σ = 0.14 tethers but excluding plectonemes, indicating FWHM of 251, 214, 205 nm for σ = −0.28, 0, 0.14 respectively (s.d. error 0.2%, "Methods" section). **e** One representative tether from fluorescence microscopy, low force plectoneme formation, anchor/magnetic bead indicated with green/magenta circles: when DNA overtwisted a plectoneme forms/grows (inset, with altered display settings to match local background). Removing positive supercoiling allows DNA to relax,

extend to original length, and removes plectoneme. After negative supercoiling applied, extension decreases and plectoneme is formed (inset, Supplementary Movie 6). **f** One representative example of longer 48.5 kbp λDNA tether, ~70% fractional extension (~0.3 pN), more plectonemes form upon torsional manipulation, three indicated (small circles); single-strand nick forms at ~860 ms (relative to start of fluorescence acquisition) causing torsional relaxation wave which diffuses along tether to remove plectoneme positive supercoiling which disappear sequentially. **g** Rate of plectoneme diffusion here decreases with increasing bp content. Diffusion coefficient presented as linear regression of mean square displacement ± a 95% confidence interval, plectoneme size as mean values ± s.e.m. over 41 imaging frames, from one DNA tether. **h** Integrated plectoneme intensity remain stable before nick (no net bp turnover), though qualitative evidence of anticorrelation between cyan/blues traces whose plectonemes separated by a few microns. **g, h** same colour code as circled plectonemes in (**f**). Experiments in panels (**c, e** and **f**) each performed once.

Conversely, plectonemes under the same conditions are predicted to form at the DIG-functionalised end of the DNA, opposite to that observed experimentally (Fig. 6c), although the plectoneme prediction algorithm was developed for exclusively positively supercoiled

DNA and we are not aware of studies using it for negative supercoiling. The analysis predicted that plectoneme formation during positive supercoiling was most likely located at the midpoint of the DNA tether, consistent with experimental observations of puncta (Fig. 6c).

**Table 1 | Plectoneme mobility is supercoiling-dependent**

| σ | rmsd$_\parallel$ (nm) | rmsd$_\perp$ (nm) | rmsd$_{tot}$ (nm) | $D_{tot}$ (nm²s⁻¹) |
|---|---|---|---|---|
| 0.14 | 108 ± 37 | 96 ± 58 | 144 ± 47 | 1900 ± 1200 |
| 0.11 | 76 ± 52 | 72 ± 50 | 105 ± 41 | 990 ± 770 |
| −0.14 | 49 ± 35 | 37 ± 14 | 62 ± 28 | 310 ± 280 |

1-dimensional root mean square displacements parallel (rmsd$_\parallel$) and perpendicular to tether (rmsd$_\perp$), and combined total 2-dimensional rmds (rms$_{tot}$) and diffusion coefficient $D$ for the puncta shown in Fig. 5e. The perpendicular thermal motion of the same tether without applied supercoiling was found to be 66 ± 46 nm, s.d. errors.

Formation of a localised bubble is a precursor of dsDNA melting, leaving two regions of single-stranded DNA (ssDNA) with high local flexibility which can form the tip of a plectoneme, a scenario which is energetically and entropically favourable compared to forming a second bubble. We therefore hypothesise that bubbles generated through low negative supercoiling act as subsequent nucleation points for plectonemes as negative supercoiling increases. SYBR Gold is capable of binding to ssDNA in addition to dsDNA, with the result that during the sampling time used experimentally, rapidly fluctuating ssDNA conformations will result in localised image blur, which is what we observed experimentally (Fig. 5d). However, fluorescence imaging of cy3b-SSB ("Methods" section), which indicated binding to ssDNA oligos, showed no binding at comparable levels in tethers, which may mean that the typical bubble size is smaller than the probe ~50 nt footprint[56]; although SIDD modelling indicated bubble sizes of ~60 nt these are limited by not considering plectoneme dependence, which may result in smaller effective bubble sizes. For positive supercoiling, considering only the intrinsic curvature and sequence accurately predicted the plectoneme location, as described previously.

To better understand bubble/plectoneme coupling, we performed MD simulations of DNA fragments held at constant force with varying σ (Fig. 6d, e, "Methods" section and Supplementary Movies 13–16). We found that plectoneme formation was always accompanied by bubbles forming at or near plectoneme tips, caused by significant bending in this region, while relaxed DNA showed no evidence for bubble or plectoneme formation. However, the nature of these bubbles varied between over- and undertwisted fragments. For overtwisted, bubbles remained relatively close to a canonical conformation, with bases on opposite strands still pointing towards each other. For undertwisted, the structural damage was catastrophic; base pairs were melted and bases face away from the helical axis, leaving large ssDNA regions. In addition, bubbles in undertwisted DNA tended to be longer (up to 12 bp) and more stable (lasting for up to 2200 ns in our simulations, though since simulations were performed in implicit solvent there is no direct mapping to experimental timescales) than the ones formed in overtwisted DNA (maximum of 5 bp and 100 ns) resulting in a significantly higher prevalence of bubbles for negatively supercoiled vs positively supercoiled DNA (Fig. 6d and Supplementary Fig. 15). This agrees with previous experimental studies: while the presence of defects has been widely reported in negatively supercoiled DNA, they have only been detected recently in strongly positive supercoiling with σ > 0.06[57]. We hypothesise that the differing nature, size and frequency of these bubbles accounts for the relative mobilities of plectonemes we observe; for the less disturbed positively supercoiled plectonemes, motions away from bubbles have a lower energy barrier than for undertwisted plectonemes which would need to form a large highly disrupted site during motion.

## Discussion

Three existing approaches of molecular manipulation correlated with fluorescence visualisation enable DNA supercoiling to be studied: (1) utilising stochastic biotin-avidin unbinding[26] between an optically trapped bead and DNA molecule, followed by transient torsional relaxation and religation which introduces negative supercoils.

Fluorescence microscopy is applied to visualise DNA. The approach is valuable in not requiring engineering of precise magnetic tweezers as with COMBI-Tweez, however, the introduction of supercoils is stochastic and limited to negative superhelical density, whilst COMBI-Tweez enables more scope to explore positive and negative supercoiling dependence on DNA topology with higher reproducibility; (2) permanent magnetic tweezers[22] in a vertical geometry to twist a magnetic bead conjugated to the free end of surface-tethered dye-labelled DNA to form plectonemes through positive supercoiling, followed by orthogonal rotation of magnets to create an obliquely oriented tether which is almost but not entirely parallel to the microscope focal plane due to imprecision with knowing the exact position of the surface relative to the bead, but which can enable plectoneme dynamics visualisation via epifluorescence. This method reports visualisation of mechanically induced plectonemes though not visualisation of mechanically controlled bubbles nor of mechanically induced plectonemes directly correlated in real-time to the mechanical driving torque; it does not require accurate engineering of coils nor optical tweezers, however, there is no high-precision force measurement and control compared to COMBI-Tweez, there are potential issues with surface interference and uncertainty in the angle between tether and coverslip leading to imprecision concerning plectoneme mobility and, importantly, it does not enable observation of plectoneme formation at the same time as twisting DNA, so it is not possible to study early onset plectoneme dynamics but only several seconds following mechanical perturbations; (3) intercalation of Sytox Orange dye[23] by surface-tethered DNA induces torsional stress which is resolved by plectoneme formation through positive supercoils, but under certain concentration regimes can also enable negative supercoils to be added. This approach is relatively easy to configure, allows for simultaneous visualisation of tethers and plectonemes, and can be performed with higher throughput than COMBI-Tweez. However, a key advantage of COMBI-Tweez is that it can control the state of DNA supercoiling directly, reversibly and consistently without relying on stochastic intercalating dye binding/unbinding which, unavoidably, varies from tether to tether. COMBI-Tweez is also able to precisely exert biologically relevant force regimes to DNA whilst observing plectoneme formation and dynamics. The independent control of torque and force over a physiological range with high-speed data acquisition coupled to high-precision laser interferometry is also a specific capability which can enable biological insights in real-time structural dynamics of DNA and processes that affect its topology over a more rapid timescale than is possible with existing approaches, such as DNA replication, repair and transcription. Also, COMBI-Tweez can monitor how torsional dynamics affects plectonemes in real-time, and can quantify torsional interactions between two DNA molecules to high-precision, which has not to our knowledge been reported with these earlier technologies. Being based around a standard inverted microscope, it is cost-effective to implement offering access to interchangeable widefield and narrowfield illumination modes. These modes can be multiplexed with trivial engineering adaptations to enable multicolour excitation or combined with techniques such as polarisation microscopy[25,58,59] to extract multidimensional data concerning biopolymer molecular conformations.

Although a previous report indicated the single-molecule fluorescence imaging of plectonemes in DNA using intercalators to change σ with both DNA ends attached to a coverslip[23], we report here to our knowledge the first transverse single-molecule fluorescence images of mechanically controlled supercoiling-induced higher-order DNA structural motifs including both plectonemes and bubbles. In the same earlier report it was indicated that plectonemes had high mobility equivalent to a diffusion coefficient of ~0.13 μm²s⁻¹, which contrasts our observation using COMBI-Tweez of a lower mobility equivalent at its highest to ~0.002 μm²s⁻¹ for the shorter 15 kbp construct, but is comparable to that observed for full length λ DNA. Aside from these length

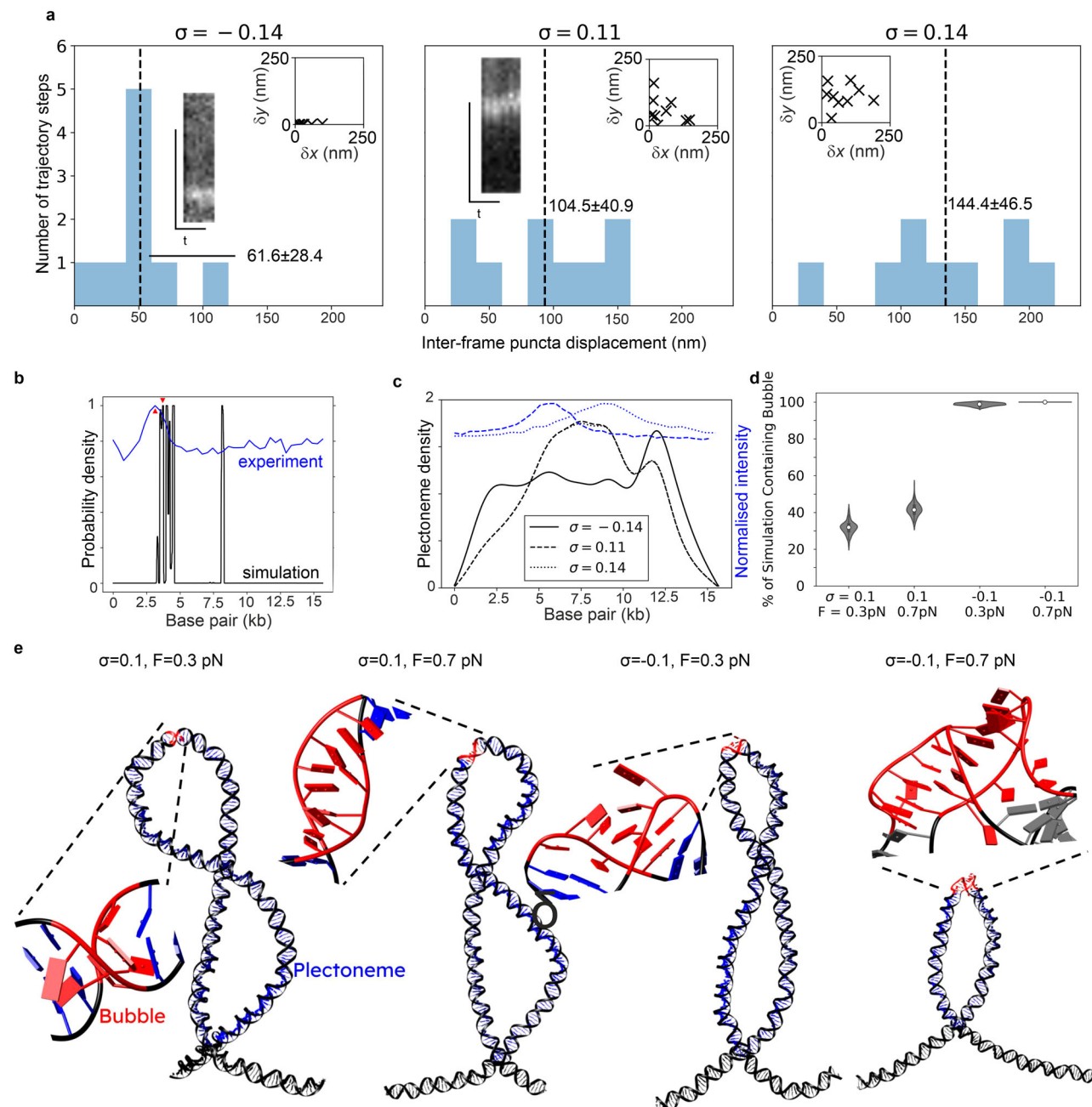

**Fig. 6 | Plectonemes are mobile but form at preferred locations. a** Distribution of inter-frame displacements for puncta mobility as assessed by 2D Gaussian tracking and calculating frame-to-frame displacement for tether with σ = −0.14, 0.11, and 0.14 respectively. Inset: scatter plots of frame-to-frame displacement in parallel to (y) and perpendicular to (x) the tether in nm, kymographs taken from a profile along the centre of the tether axis. **b** Bubble forming probability density function as predicted by Twist-DNA[78], σ = −0.14. Plotted in blue on same axes is the normalised, background-corrected intensity of the plectoneme imaged from a single tether in fluorescence microscopy (Fig. 4c). Red triangles indicate the closest agreement between experimental and predicted peaks (within 3.6%). **c** Plectoneme formation probability as predicted by the probabilistic model of Kim et al.[53] with modified average plectoneme sizes. Solid line: −200 turns (σ = −0.14), dashed line: +150 turns (σ = 0.11), dotted line: +200 turns (σ = 0.14). Dotted and dashed line overlap so they cannot be distinguished in the plot. **d** Violin plot indicating the percentage of each simulation over which bubbles are seen, determined using *n* = 200 separate bootstraps for each dataset ("Methods" section), median (white circle), interquartile range (thick line) and 1.5× interquartile range (thin line) indicated. Lack of error in σ = −0.1, F = 0.7 pN case indicates presence of at least one bubble in every frame. **e** Representative structures of our simulations highlighting plectonemes in blue and denaturation bubbles in red.

differences, there are also differences in intercalating dye used, in that the previous study used Sytox Orange dyes, whereas we focus on SYBR Gold. We speculate that the different observed diffusion response may also result from the interplay between applied supercoiling, axial force on differently sized plectonemes. The plectonemes reported previously were relatively small, equivalent to ~0.15–4 kbp, and were generated by relatively small supercoiling changes of σ ~ −0.025 at

forces typically <0.5 pN. Conversely, in our study we used a higher σ and forces closer to the physiological scale, resulting in larger plectonemes which we find does affect their mobility, since it is known that DNA in general responds to external mechanical stresses by preserving stretches of B-DNA while other regions are highly distorted[19,58]. Also, as our previous MD simulations of the emergence of structural motifs in DNA in response to stretch and twist indicates[31], we find that sequence

differences make significant impacts to DNA topological responses to mechanical stress and damage.

The model we applied involves the SIDD algorithm[54] which is limited in its ability to accurately predict where bubbles form since (1) it assumes torsional stress is partitioned into twist, effectively that the DNA is fully extended, therefore over-predicting bubble prevalence if plectonemes are present; (2) it does not consider DNA curvature on plectoneme or bubble formation. With these caveats, our calculations perform well in the context of single-molecule experiments and previous modelling studies in which we saw excellent agreement to AFM data[34,60] for bending angles and radii of gyration in addition to validation against bulk elastic properties of DNA demonstrated by the SerraNA algorithm we reported previously[61].

The agreement between predictions of bubbles and plectonemes with the observed positions of puncta for negatively supercoiled DNA raises several questions. Firstly, do bubbles nucleate plectonemes? This would be entropically and energetically favourable compared to forming two bubbles – one at the melting site and one at the plectoneme tip. Previous studies indicate a balance between denatured and B-DNA that depends on torsional stress[19,62], and utilising a pre-existing bubble to nucleate plectonemes drives this balance towards more B-DNA, that may be physiologically valuable in maintaining overall structural integrity. Secondly, is the smaller bp content measured for puncta during undertwisting compared to overtwisting a consequence of the seeding of plectoneme nucleation by a bubble? Future studies using dyes sensitive to both ssDNA and dsDNA such as Acridine Orange[63] may delineate the kinetics of bubble from plectoneme formation, since if initial bubble formation takes a comparable time to form compared to subsequent plectoneme nucleation this may explain the time-resolved differences observed in plectoneme content between under- and overtwisted DNA. Thirdly, can a bubble act as a "staple", providing an entropic and energetic barrier to plectoneme diffusion, holding it in place at the plectoneme tip – as mismatched/unpaired regions have been shown to do in experiment[23] and simulation[30]? If so, since predictions indicate sequence-dependence on bubble formation, could plectoneme formation serve a role to regulate DNA topology at "programmed" sites in the genome? Given that in *E. coli* the genome is negatively supercoiled with a mean σ ~ −0.05 could sequence repeats with enhanced likelihoods for bubble formation serve as a hub for plectonemes that prevent mechanical signal propagation along DNA, in effect defining genetic endpoints of "topological domains" that have roles in protein binding and gene expression[57]?

To our knowledge, this is the first time that size, position, and mobility of higher-order structural motifs in DNA under mechanical control have been *directly* measured in real-time simultaneously with the driving mechanical perturbation. The dependence of DNA shape and mechanics on its interactions with its local environment represents a divergence from traditional views of DNA seen through the lens of the Central Dogma of Molecular Biology; COMBI-Tweez has clear potential to facilitate mechanistic studies of DNA topology dependence on the interactions with binding partners such as enzymes, transcription factors and other nucleic acid strands. And, with trivial adaption, the technology can be implemented to study other filamentous chiral biopolymers; RNA is an obvious candidate, but also modular proteins such as silk show evidence for interesting torsional properties[64] which have yet to be explored at the single-molecule level in regards to their response to twist. It may also be valuable to use COMBI-Tweez to explore specific mechanistic questions relating to emergent features of DNA topology, including the dependence on force and ionic strength on DNA buckling, to further probe the source of very rapid fluctuations we observe in DNA supercoiling using bfp detection; the ~2-fold increase in fluctuation amplitude with increasing superhelical density during overtwist may signify transitions between metastable structures of relatively small DNA segments

which simulations previously indicated emerge during extension of torsionally-constrained DNA in a sequence-dependent manner[31]. Similarly, our observation of an increase in the lower frequency power spectral components of buckling DNA merits further investigation to study whether this effect is influenced by sequence, salt and plectoneme formation. Also, there may be value in using this instrumentation to investigate long-range rapid plectoneme hopping mobility reported previously[22], as well as probing the role of DNA topology in crucial cell processes involving interaction with DNA binding proteins, such as DNA repair mechanisms[65]. The collation of single-molecule tools in COMBI-Tweez around a single optical microscope also presents valuable future opportunities to integrate even more single-molecule biophysics tools, many of which are developed around optical microscopy[66–68].

## Methods

### Instrumentation

Optical tweezers (Supplementary Fig. 1) were built around an inverted microscope (Nikon Eclipse Ti-S, Nikon Instruments Inc.) with NIR trapping laser (opus 1064, Laser Quantum), to overfill objective aperture (100×, NA 1.45, oil immersion, MRD01095, Nikon Instruments Inc.). Oil immersion condenser (NA 1.4, Nikon Instruments Inc.) recollimated beam, back focal plane imaged onto a QPD[43] (QP50-6-18u-SD2, First Sensor) and digitized (NI 9222 and NI cDAQ-9174). For most fluorescence microscopy, a 488 nm wavelength laser provided excitation, detection via a −80 °C air-cooled EMCCD (Prime 95B, Photometrics, or iXon Ultra 897, Andor Technology Ltd, 55 and 69 nm/pixel respectively). Magnetic tweezers were generated using Helmholtz coils (Supplementary Fig. 2) on an aluminium platform. B field measurements indicated uniformity over several mm (maximum DC force $4.2 \times 10^{-5}$ pN, Supplementary Note 1). SWG20 copper wires (05-0240, Rapid Electronics Ltd.) were wound onto 3D printed spools (Autodesk Inventor, Object30, material: VeroWhitePlus RGD835), 95 turns for two smaller spools, 100 turns for two larger spools[69]. LabVIEW signals were generated (NI 9263 and NI cDAQ-9174); two bipolar 4-quadrant linear operational amplifiers (BOP 20−5 M, Kepco, Inc.) received voltages to convert to coil currents. COMBI-Tweez was optical table mounted (PTQ51504, Thorlabs Inc.) in aluminium walls/card lids, ±0.1 °C climate controlled with air conditioning (MFZ-KA50VA, Mitsubishi Electric). Custom LABVIEW software (Supplementary Fig. 3) enabled instrument control/data acquisition. Phased/modulated currents were sent to generate a 2D rotating B field; QPD (2 axes), current (2 coils), nanostage (3 axes) and camera fire signal were sampled at 50 kHz.

### Characterising tweezers

40 nm beads were incubated with 3 μm magnetic beads to impart morphological asymmetry detectable on the QPD as a magnetic bead rotated. A 0.54 A sinusoidal 1–8 Hz current was sent to the larger coil pair, 1.5 A to the smaller pair (Supplementary Fig. 5a). We evaluated phase shifts between coil input and bead rotation by taking QPD voltage correlated with its time-reversed signal to reduce noise for peak detection. Supplementary Fig. 5b plots phase shift as peak vs rotation frequency. Data were fitted using linear regression to yield gradient −0.429 rad Hz$^{-1}$, angular stiffness $k_\theta = 8\pi\eta R^3/|\text{gradient}| = 1.1 \times 10^3$ pN · nm · rad$^{-1}$. To confirm no OT/MT interdependence we intermittently switched a rotating 1 Hz B field on/off for 10 s intervals and monitored QPD signals in the absence of beads, confirming no 1 Hz peak (Supplementary Fig. 5c). We could entirely remove any driving 1 Hz signal from QPD responses of rotating trapped beads using narrow bandpass filtering, indicating no intrinsic influence of bead rotation on QPD signals.

We measured trapping force as displacement of a magnetic bead from the trap centre (estimated from mean-filtered QPD signals[42], using prior 2D raster scanning of surface-immobilised beads as calibration) multiplied by trap stiffness (determined from corner

frequency of a Lorentzian fitted to the trapped bead power spectrum), Supplementary Fig. 5.

## Force clamping

A Proportional Integral Derivative (PID) feedback loop was enabled between QPD and nanostage to keep tether force constant parallel to its axis (defined as $x$) by dynamically repositioning the nanostage by the difference between set and measured force divided by trap stiffness (10 pN/µm), using PID optimisation to prevent overshooting/discontinuities. The clamp response time was ~1 s (Supplementary Fig. 6) with set force constant ±0.1 pN throughout entire over-/undertwisting experiments (Supplementary Fig. 7). Small forces detected parallel to $y$- and $z$-axes due to angular constraints in trapped beads were minimised by manual nanostage adjustment.

## DNA preparation

A ~15 kbp DNA construct was synthesised by ligating functionalized handles, containing biotin-16-dUTP or digoxigenin-11-dUTP (Roche, 1093070910/11277065910 respectively), to a λ DNA fragment (Supplementary Fig. 8). Two PCR reactions were performed using primers NgoMIV forward/NheI reverse, amplifying a 498 bp region of plasmid pBS(KS + ) generating a 515 bp handle. For biotinylation, the ratio of dTTP to biotin-16-dUTP was 1:1 generating ~120 biotin-16-dUTP per handle. For digoxigenin-labelling, the ratio of dTTP to digoxigenin-11-dUTP was 6.5:1 generating ~32 digoxigenin-11-dUTPs per handle. PCR reactions were monitored by gel electrophoresis and handles purified using a Qiaquick PCR kit before digestion with NgoMIV (biotinylation) or NheI-HF (digoxigenin-labelling). The remaining tether comprises a 14.6 kb fragment excised from λ using NgoMIV and NheI-HF, purified using the Monarch gel extraction kit from a 0.5% agarose TBE gel run at 2 V/cm for 36 h. Handles were mixed with cut λ in 5:1 handle:λ ratio (two compatible sticky ends for each handle), ligated using T4 DNA ligase (NEB). 48.5 kbp λ DNA was also created using a previously reported protocol[26], removing phosphorylation steps using T4 polynucleotide kinase as already supplier-phosphorylated (IDT N3011S).

## Bead functionalisation

To suppress bead brightness in fluorescence (Supplementary Fig. 9), beads were functionalised; 250 µL of 50 mg/mL solution of Micromer or Micromer-m (Micromod 01-02-503/08-55-303 respectively), were incubated with 62.5 µL of 5× MES (Alfa Aesar J61587), 2 mg of EDC Hydrochloride (Fluorochem 024810-25 G), and 4 mg N-Hydroxysuccinimide (Sigma 130672-5 G) at room temperature for 45 min while vortexing. Beads were pelleted by centrifugation at 12,000 × g and resuspended in 200 µL of 200 µg/mL anti-digoxigenin (Merck Life Sciences 11333089001) or 200 µL of 200 µg/mL NeutrAvidin (Thermo Scientific 31000), then further incubated 3 h. Beads were pelleted and resuspended in 100 µL PBS and 25 mM glycine to quench for 30 min. Beads were centrifuged/resuspended 3× in 500 µL PBS before resuspending in 250 µL PBS/ 0.02% azide, generating anti-digoxigenin Micromer beads and NeutrAvidin Micromer-M beads, or the converse at 50 mg/mL.

## Sample preparation/tethering

A flow cell was prepared using truncated slides by scoring/snapping a 50 × 26 mm glass slide (631-0114, VWR), forming a double-sided tape plus 22 × 22mm coverslip "tunnel" as described previously[70]. Flow cells were nominally passivated using polyethylene glycol (PEG)[71] variants MeO-PEG-NHS and Biotin-PEG-NHS (Iris biotech PEG1165 and PEG1057 respectively). Anchor and trapping beads were commercially carboxylated then functionalized with anti-digoxigenin or NeutrAvidin as above. 5 µm anti-digoxigenin anchor beads were diluted to 5 mg/mL in PBS/0.02% sodium azide and 2 mM MgCl2 and vortexed to disaggregate. 10 µL was introduced, inverted and incubated in a humidified chamber 15 min room temperature for surface immobilisation;

for PEG passivated flow cells 5 µm NeutrAvidin anchor beads were used. 20 µL 2 mg/mL BSA in PBS was introduced and incubated 5 min. Nominally, 10 µL DNA 0.12 ng/µL in T4 DNA ligase buffer and with 1 µL T4 DNA ligase was introduced and incubated 1 h, washed 200 µL PBS, 20 µL imaging buffer introduced, and flow cell was sealed with nail varnish.

For stretch-release experiments we used ~1 Hz nanostage triangular waves, amplitude ~5 µm, acquiring for ≥2 consecutive cycles. For fluorescence, the imaging buffer comprised PBS, 1 mg/mL 3 µm NeutrAvidin-functionalised Micromer-m beads, 6% glucose, 1 mM Trolox, 833 ng/mL glucose oxidase, 166 ng/mL catalase, 25 nM SYBR Gold. For brightfield-only experiments, the imaging buffer comprised 1 mg/mL 3 µm NeutrAvidin-functionalised Micromer-m beads in PBS. Fluorescence experiments used 40 ms exposure time, maximum camera gain. Oblique-angle Slimfield[66] 0.11 kW/cm² was the default mode, but we confirmed compatibility with epifluorescence, TIRF and HILO. We first imaged DNA for a few seconds at 0.1 mW to enable focusing but avoiding photobleaching/photodamage, then imaged at 1 mW. For continuous imaging, we acquired data for 200 rotations. For discontinuous imaging, we recorded 50 bead rotations using no fluorescence, paused 1 min, then10 frames in fluorescence, acquiring ~10 supercoiling states per molecule. Brightfield-only experiments were 40 ms exposure time, zero camera gain, no camera cooling. σ was calculated as number of bead rotations divided by number of turns in relaxed DNA (B-DNA twist per base pair multiplied by number of bp in the DNA construct).

An anchor bead was selected and nearby free magnetic bead optically trapped. Beads were brought within 500 nm and incubated 2 min to facilitate tether formation (optimised using fluorescence by acquiring 10 frames to visualise DNA and reposition beads). Beads were separated ~5 µm and visualised with fluorescence if appropriate to test if a tether had formed (Supplementary Movies 2, 6). A force-extension curve was generated by oscillating the trapped bead and fitted by a wormlike chain to generate persistence and contour lengths. Throughput, dependent on DNA concentration, was nominally one in ~20 attempts (~5 min) using 0.24 ng/µL, analysis indicating binding fraction for >1 tether was 5%[72]. Using 0.48 ng/µL enabled controllable formation of two tethers between bead pairs to explore the effects of braided DNA. Surface drift was quantified from brightfield video-tracked intensity centroid displacements every 30 s up to 1 h of surface-immobilised beads in the absence of tethering, indicating mean of ~15 nm/s and ~7 nm/s for lateral and axial drift respectively. Trapping drift was assessed using QPD signals from a trapped untethered magnetic bead but aside from expected rms fluctuations of a few tens of nm no directed drift was detected.

## Imaging SSB/hRPA

Surface-immobilized 100nt ssDNA-cy5 and ssDNA binding proteins (cy3b-SSB[73] and hRPA-eGFP[74]) were imaged on PEG passivated slides, incubated 5 min at room temperature with 200 µg/mL Neutravidin (Thermofisher Scientific, 31000) in PBS followed by 200 µL wash. 10 µL 500pM ssDNA-cy5 in PBS was introduced and incubated 10 min to allow binding of 5' biotin on ssDNA-cy5 to Neutravidin on flow cell surface, excess ssDNA-cy5 washed 2 × 100 µL PBS. 100 µL 100 nM cy3b-SSB/hRPA-eGFP in PBS was introduced before the flow cell was sealed. A bespoke single-molecule TIRF microscope with ~100 nm penetration depth[75] imaged surface-immobilised binding complexes, excitation was by an Obis LS 50 mW 488 nm wavelength laser (hRPA-eGFP); Obis LS 50 mW 561 nm wavelength laser (cy3b-SSB); Obis LX 50 mW 640 nm wavelength laser (ssDNA-cy5), 10 mW at 14 µW/µm². Both probes showed colocalization when incubated with surface-immobilised ssDNA-cy5 oligo. Since cy3b-SSB was more photostable and less impaired by steric hindrance than hRPA-eGFP we focused on cy3b-SSB to probe DNA tethers. Due to PEG slide passivation, anchor beads were functionalised with NeutrAvidin to bind to PEG-biotin

coverslips, so Micomer-M beads were functionalised with anti-digoxigenin. Tether preparation was as before with addition of 10 nM cy3b-SSB to the imaging buffer, and alternating 488 nm/561 nm wavelength laser excitation[76], 1 mW/10 mW respectively.

## Wormlike chain fitting

Force-extension data were fitted by a wormlike chain (WLC) using Python3 and numpy's curve_fit to minimise least squares, error estimated from bootstrapping by taking 1% of randomly selected data, WLC fitting, iterating 1000 times, indicating ~0.2% s.d.

## Puncta analysis

To assess percentage of tether bp in a plectoneme we used ImageJ to integrate background-corrected pixel intensities associated with puncta normalised to the integrated background-corrected intensity for the whole tether. To measure two-dimensional rms puncta displacements, we used single-particle tracking software PySTACHIO[77] with mean square displacement $<r^2> = 4D.\delta t$ where $D$ is the two-dimensional diffusion coefficient, and $\delta t$ the inter-frame time interval 40 ms, to 20 nm localisation precision.

## Plectoneme/bubble calculations

Predictions for plectoneme loci were made using a model[53] considering DNA curvature as determining factor, modified to increase the largest possible plectoneme to deal with the ~15 kbp construct and higher σ levels. Cutoff was increased from 1 kbp, as set in the original model, to 8.3 kbp for σ = 0.14 (or +200 DNA turns), to 6.7 kbp for σ = 0.11 (+150 DNA turns) and to 1.9 kbp for σ = −0.14 (or −200 DNA turns); limits were differently established according to experimental estimations of number of plectoneme bp. Predictions for bubble loci were made using the Stress-Induced DNA Destabilization (SIDD) algorithm[54] implemented on the Twist-DNA program to deal with long sequences at genomic scales[78]. Calculations were done at σ = −0.14, 0.1 M salt, T = 310 K. For comparison of experimental fluorescence data to bubble predictions, the plectoneme tether line profile was normalised and fitted with a cubic spline, regularised and plotted on the central 14.6 kbp.

## Control of tension and torsion in silico

DNA was modelled under restraints to control tension and torsion mimicking experimental conditions (Fig. 5) using positional and 'NMR' restraints of AMBER[79]. One end of the duplex was fixed by restraining coordinates of O3′ and O5′ atoms of the final bp ('fixed end'). The other ('mobile end') was kept at force 0.3 or 0.7 pN by applying a linear distance restraint to each strand, between O3′ or O5′ of the final bp and two fixed dummy atoms[80] used as reference points (A and B, Supplementary Fig. 12). Angular restraints were applied for confining 'mobile end' motion to the tether axis (Supplementary Fig. 12, denoted z). To prevent y movement, angles $\theta_1$ and $\theta_2$ were constrained. Another pair of reference points and restraints were implemented to prevent x movement. Supercoiling was relaxed by passing DNA over either end of the molecule. To prevent 'untying', we applied one angular restraint per phosphorus atom of the bulk of the chain and per end (Supplementary Fig. 12b). Specifically, ψ angles (defined by a phosphorus atom, the last mobile O3′ or O5′ on that strand and the corresponding reference point) were forced >90°, creating an excluded volume resembling a bead. To ensure excluded volume restraints were triggered as infrequently as possible, we added 60 GC bp to the 'mobile end' as a buffering molecular stretch which were prevented from bending by dihedral restraints applied to each complementary pair of phosphorus atoms (Supplementary Fig. 12c). DNA torsional stress was maintained by ensuring O3′ and O5′ of the first relevant bp of the 'mobile end' (61st bp) were co-planar with respective reference points A and B, achieved through a dihedral restraint that reduced the angle between ABF and BAE planes to zero.

The 'fixed end' was torsionally-constrained using an equivalent dihedral angle defined by O3′ and O5′ of the next-to-last bp and reference points I and J.

## DNA in silico

The structure of a linear 300 bp DNA molecule was built using Amber18[79] NAB module from a randomly generated sequence with 49% AT surrounded by 2 GC bp at the 'fixed end' and 60 GC bp at the 'mobile end', 362 bp total; 300 bp sequence was:

[1]TGCAAGATTT [11]GCAACCAGGC [21]AGACTTAGCG [31]GTAGGTCCTA [41]GTGCAGCGGG [51]ACTTTTTTTC [61]TATAGTGTTT [71]GAGAGGAGGA [81]GTCGTCAGAC [91]CAGATACCTT [101]TGATGTCCTG [111]ATTGGAAGGA [121]CCGTTGGCCC [131]CCGACCCTTA [141]GACAGTGTAC [151]TCAGTTCTAT [161]AAACGAGCTA [171]TTAGATATGA [181]GATCCGTAGA [191]TTGAAAAGGG [201]TGACGGAATT [211]CGCCCGGACG [221]CAAAAGACGG [231]ACAGCTAGGT [241]ATCCTGAGCA [251]CGGTTGCGCG [261]TCCGAATCAA [271]GCTCCTCTTT [281]ACAGGCCCCG [291]GTTTCTGTTG

To determine the default twist, we performed a σ = 0 simulation. We used the 3DNA algorithm in CPPTRAJ[81] as a reference to build under- and overtwisted straight DNA, σ = ± 0.1.

## MD

Simulations were done in Amber18, performed using CUDA implementation of AMBER's pmemd. DNA molecules were implicitly solvated using a generalised Born model, salt concentration 0.2 M with GBneck2 corrections, mbondi3 Born radii set and no cutoff for better reproduction of molecular surfaces, salt bridges and solvation forces. Langevin dynamics was employed using similar temperature regulation as above with collision frequency 0.01 ps to reduce solvent viscosity[82]. The BSC1 forcefield was used to represent the DNA molecule[83]. A single simulation was calculated for each combination of force and σ (= ± 0.1, force F = 0.3, 0.7 pN), following our protocols for minimisation and equilibration. In addition, we performed a control simulation at σ = 0. We performed 40 ns re-equilibration, applying the above restraints with exception of tensile force, for allowing plectoneme formation; restraints on the canonical WC H-bonds were added to avoid premature double helix disruption and allow distributions of twist and writhe to equilibrate. Simulations were extended 500 ns–2.3 μs depending on convergence, measured by cumulative end-to-end distance over time (Supplementary Fig. 15b), calculated using a single NVIDIA Tesla V100 GPU from the local York Viking cluster at 40 ns/day.

## Determination of bubbles in silico

Melting bubbles were easily identifiable by visual inspection. To quantify and ensure we only captured significant denaturation, we assumed a bubble was formed by ≥3 consecutive bp that didn't present WC H-bonds, whose angular bp parameters (propeller twist, opening and buckle) were ≥2 s.d. from the average obtained from relaxed DNA and that disruption lasted for more than 1 ns. This information was acquired using the nastruct routine of CPPTRAJ[81]. Percentages of simulations where DNA presented bubbles were calculated considering the last 400 ns. Standard deviations were calculated using bootstrapping, sampling 1% of each simulation 200 times.

## Wax melting

A 1% w/v suspension of each of three different alkane waxes was made, warmed in a water bath 10 °C hotter than the alkane melting point and sonicated 1 min before plunge cooling in an ice bath, injected into a flow cell and incubating 20 min prior to 100 μL PBS washing, introducing 50 μg/mL magnetic beads, and sealing the flow cell. The experiment comprised selecting a surface-immobilized wax particle, trapping a nearby magnetic bead and bringing it in contact with the wax, then leaving it in position for 3 min, recording brightfield time-lapse movies.

## TEM/electron diffraction

3 μm diameter magnetic beads at initial concentration 50 mg/ml were pelleted by centrifugation $1000 \times g$ 1 min, washed in 100% Ethanol by resuspension/centrifugation, then infiltrated over 48 h in LR White resin (Agar Scientific), and subsequent polymerisation at 60 °C for 48 h. 70 nm sections were cut (Leica Ultracut UCT7 ultramicrotome and Diatome diamond knife). TEM was carried out on a JEOL 2100+, 200 kV using a 150 μm diameter condenser aperture, brightfield imaging achieved with a 120 μm objective lens aperture. Simulated electron diffraction was performed for $Fe_3O_4$ nanoparticles using a standard Al diffraction sample for calibration at camera length 25 cm$^{-1}$ compared to Selected Area Electron Diffraction (SAED) images taken from the edges of beads where nanoparticles were present. SAED was also taken from within beads and showed no crystal structure.

## Reporting summary

Further information on research design is available in the Nature Portfolio Reporting Summary linked to this article.

## Data availability

Experimental and simulation data used are publicly available at DOI:10.5281/zenodo.7786636 and DOI:10.15124/fd0eb563-9a9f-4c0d-82dc-27a4bf071660 respectively. Source Data for all graphs is available with this paper. Source data are provided with this paper.

## Code availability

All code used for instrument design/control, data acquisition/processing/analysis and figure generation directly relied on: Matlab, LabVIEW, Mathematica, Autodesk Inventor, Jupyter, Matplotlib, NumPy, Pandas, SciPy. COMBI-Tweez source code available at https://github.com/york-biophysics[84] under Creative Commons Attribution-NonCommercial-ShareAlike 4.0 International License (CC-BY-NC-SA; https://creativecommons.org/licenses/by-nc-sa/4.0/).

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

## Acknowledgements

This work was supported by the Leverhulme Trust (RPG-2017-1340, RPG-2019-156), BBSRC (BB/R001235/1, BB/W000555/1) and EPSRC (EP/N027639/1, EP/R513386/1, EP/R029407/1, EP/T022205/1). Thanks to Mark Dillingham (University of Bristol, UK) and Mauro Modesti (CRCM, France) for donation of cy3b-SSB and hRPA-eGFP respectively.

## Author contributions

M.L. conceived the study. Z.Z. developed tweezers/imaging instrumentation and control software, which in the following years of development S.G., J.S. expanded. J.A.L.H. developed DNA chemistry. J.S., J.A.L.H., and S.G. collected data which J.S. analysed. A.N. conceived theoretical molecular dynamics simulations, and M.B. performed them. C.S. developed an analytical temperature model. C.S-K, A.K. performed TEM and electron diffraction. J.S., M.L. wrote the bulk of the manuscript, with contribution, discussion and revision from all of the authors. M.L. was the project administrator.

## Competing interests

The authors declare no competing interests.
