## [Peer Review File · Nature Communications]

Reviewers' Comments:

Reviewer #1:

Remarks to the Author:

The authors have studied the coupling between the formation of denaturation bubbles in supercoiled DNA and the positions of plectonemes using a novel combination of microscopy and MD simulations, coupled to statistical mechanical calculations. I am able to comment specifically on the simulation aspects of the paper.

A comparison between the experiments and statistical mechanical calculations is made in the paragraph on pg 288. Do the authors have an idea for why there is a big peak in the denaturation probability in Fig 5b at ~ 7 kbps, which does not appear in the experiment? How well would these theoretical models be expected to perform in the context of these single molecule manipulation experiments? If possible, it would be helpful to replace the word "excellent" on line 292 with a quantitative measure of the performance of the calculations relative to the experimental data.

My understanding from the Online Methods section is that all of the atomistic MD simulations have been performed in implicit solvent. Such models can be extremely helpful for exploring conformational space rapidly in the absence of solvent friction, however, it is anecdotally believed that these models can destabilise dsDNA. Have the authors performed a control simulation to check that their simulations do not develop denaturation bubbles when the DNA is relaxed, and not subjected to torsional forces? This would provide the reader with increased confidence in the implicit solvent models. The implicit solvent models have been run for a very impressive (>2 microseconds) amount of time given the large (~ 300 bp) size of the DNA sequence. How many nanoseconds per day can be achieved on the computational resources used here, and where were the calculations performed? The restraints employed were clearly very technically challenging to implement, and I wonder how transferable these will be to other plectonemic DNAs.

One comment on the thermodynamics: my understanding is that the formation of one bubble is favourable relative to forming two specifically because of the nucleation barrier, which in DNA is the same regardless of the length of the bubble (as discussed in papers by Benham).

I enjoyed the supplementary movies showing the behaviour of the DNA in the atomistic MD simulations. I found it helpful to play these at 50% speed, otherwise the movie was too fast to see the red denatured regions.

Reviewer #2:

Remarks to the Author:

Referee report for Shepherd et.al.

In this manuscript, the authors characterize DNA plectoneme size, position and dynamics using a new integrative technique that combines fluorescence microscopy, optical tweezers (OT) and magnetic tweezers (MT) that is named as 'COMBI-Tweez'. OT is used to trap a magnetic bead to which a DNA molecule is tethered. MT is used to rotate the bead that induces twists in the DNA. The supercoiling leads to formation of plectonemes which is observed using fluorescence microscopy.

They demonstrate various features and capabilities of this new instrument and they support their observations using MD simulations and further quantify the dynamics of supercoiled DNA structures. This technique is interesting because it decouples the force and the torque hence enabling precise manipulation of the tethered molecule while sampling its response at kilohertz rates. However, this reviewer is of the opinion that this manuscript does not provide new insights into our understanding of dynamics of DNA plectonemes. The study lacks key quantitative details of the new technique and does not utilize its full potential to justify its value proposition.

It would greatly benefit the study if the authors could demonstrate the uniqueness of COMBI-Tweez by designing an experiment that employs the combination of the showcased features in a way that has not been done before.

Major concerns: (references in square brackets denote references from the main manuscript,)

1. The manuscript lacks sufficient motivation to develop the technique from a biological standpoint. The authors should clarify what measurements can be performed by COMBI-Tweez that cannot be performed using existing methods

a. Decoupling of force and torque: How is this decoupling important in the experiments demonstrated in the manuscript? What advantages does 50 kHz sampling rate provide in studying dynamics DNA pleconemes/ bubbles?

b. The reviewer is of the opinion that the data presented in this manuscript can also be generated from experimental setup used in reference [21] at much higher throughput. The authors should provide a quantitative explanation justifying the uniqueness of a COMBI-Tweez measurement.

2. Line 55:

- Here, the references [23,24] cited by the authors refer to studies which DO NOT use a torsional constraint on the DNA, neither they use optically trapped magnetic beads not permanent magnets. The authors should revise this sentence correctly according to the cited papers or cite the correct reference that this sentence can be related to.

- In case OT with magnetic beads has been used for the first time in this study, the authors should mention it explicitly in the manuscript and provide control experiments showing that the magnetic bead does not absorb significantly in the NIR wavelength used in OT. Dissipation of heat from the irradiated magnetic beads can have a considerable effect on the DNA tether or its dynamics. This effect should be quantified and demonstrated to be insignificant over the time scales involved in typical MT or OT assays.

3. COMBI-Tweez being a new technique, should be benchmarked and compared with independently existing techniques. Some key quantitative details missing from the study are:

- What is the throughput of this technique? What is the typical number of successful tethers that can be analyzed? What percent of DNA tethers show single tethers or double tethers at given experimental conditions?

4. Line 111: The authors do not provide any direct evidence of decoupling of force and torque. The use of magnetic beads in OT is not a common practice. Neither is it common to apply torsion to optically trapped magnetic beads. Control experiments should be performed, and their results added to Fig 1. or to SI. Two control experiments can be performed.

i. To show that force (OT) is not affected by torque (MT) due to rotating magnetic field- monitoring XY position of the trapped bead while rotating the magnetic field by using torsionally unconstrained DNA as a tether.

ii. To show that torque (MT) is not affected by force (OT)- repeating "Hat-curve" experiments for various OT trap stiffness (varying intensities).

These are my recommendations, however the authors can choose different control experiments to validate the decoupling.

5. Line 182, 262: The authors observe negligible hysteresis in the rotation experiments. Does this mean that the magnetic bead can torionally relax by thermal fluctuations? If so, this would lead to incorrect measurement of applied torque on the DNA molecule therefore and might be an important source of error that needs to be minimized.

6. Line 383: The authors claim to report first images of mechanically controlled bubbles: This claim is not completely valid because,

a. Mechanically induced pleconemes have been visualized before (reference [21], main text).

b. The authors observe bubbles qualitatively as a localized blur. They should quantify these blurs and predict the number to base pairs inside the bubbles. To definitively claim that the blurs are because of ssDNA in the bubbles, a fluorescently labelled molecule that binds only to ssDNA (for.eg. RPA- EGFP in ref. [23], main text) should be used to specifically image ssDNA bubbles.

Minor Comments:

1. There is very little reference to the relevant sections of online methods throughout the main text. The authors should refer to related sections of online methods and SI more frequently in the main manuscript.

2. LINE 76: The authors claim that conventional MT introduces mechanical vibrations and limits the timescales that can be probed using MT. The authors should specify an upper bound for these timescales and compare it to the fastest time scales that have been probed in this study.

3. Line 150: What does lower OT stiffness refer to in the context of this study? The authors should specify the values of OT trap stiffness quantitatively and compare them with typical OT values used in similar experiments.

4. Line 212: The manuscript presents stretching/twisting data from only 9 tethers. I find this number to be significantly lower to produce statistically significant data shown in Table 1. Is this number lower due to some experimental challenges? If yes, they should be mentioned.
5. Line 246, "... a constant force above F_c to a SYBR Gold 247 labelled tether...". Here, what is the value of F_c ?
6. Line 162 and Figure 2b. At point C, the bead exits the OT. In the figure 2b. one can see the tether length jump to ~ 5.6 μm however in the inset, the bead is closer to the surface immobilized bead. Is the jump in the displacement to 5.6 μm at point C an artifact of the measurement? If so, it should be mentioned in the figure captions.
7. Figure 2C.
 - a. There are no arrows indicating the path of the experiment in the figure like the authors have mentioned in the captions.
 - b. There are two traces (four independent lines) in the plot. The authors should clarify what they mean or show them in different color with appropriate legend in the plot.
8. Line 234-235: How does observed 25% increase in contour length explain the 0.5 stoichiometry of the fluorescent dye? The authors should provide a brief justification for this observation or the reference [37] should be cited at the end of the sentence.
9. Figure 3 caption;
 - a. 3.c) Does $n=4$ refer to 4 stretch release cycle of the same tether? Or 1 cycle for 4 different tethers? In either case, is this number low because of the experimental constraints? In my opinion performing 30-50 stretch-release cycles instead of bootstrapping the data from the same cycle should be more meaningful.
 - b. Even if the bootstrapping procedure gives negligible standard deviation, the authors should still include it in the manuscript. This is because in the jitter plots of contour length and the persistence length, they do observe a significant variance in the values calculated by the same procedure.
 - c. The authors should point to the relevant sections in the online methods as they introduce new experimental or analytical technique.
10. Figure 4.a : arrows indicating the direction or a legend referring to the colors of the trace should be added to the figure. Such color scheme should also be implemented for figure 2.c
11. Figure 4c and d: the panels could be independently labelled (for.eg. with roman numerals) for more clarity and accurate referencing in the main text.
 - a. Figure 4.d, left most panel has a misplaced letter "c" in the image. This should be corrected.
 - b. Figure 4.d. The scale bar is different in different panels. I recommend making it uniform across all the panels.
12. Figure 5.d. Predictions of the simulation is that 40% of the positively supercoiled DNA show bubbles. This number is not 100% as in the case of negative supercoils but its significantly higher and hence demands an explanation.
13. SI: Figure 1. There appears to be a blue box in the top right corner which seems to be there by mistake.
14. SI: figure 6. Caption (Line 97) misspelled "instaneously" .

Reviewer #3:

Remarks to the Author:

In the manuscript "DNA plectoneme size, position, and mobility revealed by fluorescence microscopy and combined optical and magnetic tweezers", the authors describe the construction of a novel microscope that combines fluorescence, magnetic tweezers, and optical tweezers into a single instrument. The design seems to nicely decouple the applied torque from applied forces, allowing for high bandwidth measurements of force and extension at a given degree of supercoiling. They then attempt to characterize the performance of the new microscope by measuring properties of double stranded DNA. Finally, they demonstrate the use of all three techniques to simultaneously measure plectoneme formation in supercoiled DNA. While the microscope is technically impressive and has the potential to be useful, the current manuscript suggests there are numerous bugs that need to be characterized before it is ready. These include concerns over heating of the magnetic bead by the optical tweezers, reproducibility of the optical tweezer measurements, a high degree of drift in the magnetic tweezers, and odd discontinuities in the "hat curves". They also do not offer a clear experiment that could only be done using this instrument, and the experiments they do perform are often done with only a single example,

making it impossible to verify the reproducibility of the results. Because of these major concerns, I would not recommend publishing the current manuscript.

MAJOR CONCERNS

(1) Heating of the bead:

Typically optical tweezers will be used to trap polystyrene or glass beads that do not absorb much NIR light. The metal core of magnetic beads can absorb much more IR light, raising the possibility of heating the sample appreciably. The authors seem to have encountered this problem, since they mention that they “configured the OT stiffness to be relatively low compared to earlier studies of DNA that probed the over-stretch transition, allowing stable trapping over a range of a few pN relevant to the physiological scale while avoiding issues of localized heating of the trapped magnetic bead due to NIR laser absorption.” This heating issue has been a hurdle to other attempts to combine magnetic and optical tweezers, so the authors should specify how much heat is being created by the tweezers. Does this limit the maximum force that can be applied? Would it effect the kinetics of enzymes? Given the change in viscosity of water with temperature, measurements of the power spectrum should allow the authors to measure the amount of heating occurring near the sample.

(2) Reproducibility of OT:

The measured persistence lengths of the DNA shown in Fig. 2a vary by nearly an order of magnitude. In a typical optical tweezer instrument this would be considered far too high for reliable measurements. The authors make no attempt to explain this high variation. It could be caused by errors in the stiffness of the bead, drift in the position signal, or issues with the geometry of the setup, but some discussion of why this error is so high is needed to understand what other measurements would have the same high uncertainty. Also troubling is the fact that in Extended Data Fig. 1 there is a force extension curve where the DNA apparently gets shorter as the tension is increased. This may indicate drift is too high of a factor to even complete a force-extension curve.

(3) Apparent drift of magnetic tweezers:

The “hat curves” shown in Fig. 2c do not resemble typical hat curves on DNA. They exhibit far more hysteresis than has been reported previously, and as the superhelical density approaches zero they do not match up on the left- and right-hand sides. There seems to be drift in either the measured extension or the measured force that is biasing these results. Discussion is needed to characterize the source of these anomalies so readers can evaluate what other measurements could be affected. The authors also need to repeat the measurements on more than one DNA strand so the stability of the measured quantities can be evaluated.

(4) Strange discontinuities in magnetic tweezer data:

Both Fig. 2c and Extended Data Fig. 2 show a sharp corner as the buckling transition for plectoneme formation kicks in. This could indicate a potential use of the bandwidth of this instrument in study dynamics near this point, since the data seems to resolve the process in finer detail than a typical magnetic tweezers setup. However, Fig. 4a also shows sharp corner-like discontinuities at seemingly random locations. These bumps in the data make it hard to interpret what might be happening at the buckling transition.

(5) Final experiment fails to emphasize microscope’s abilities:

The paper should include an example of an experiment that could only be done on the new instrument do demonstrate its usefulness. Unfortunately, the experiments described in Fig. 5 could also be performed on other instruments (e.g. the apparatus described in Van Loenhout et al. (ref. 21), which used magnetic tweezers to both twist and apply force to a fluorescently labeled DNA molecule). An easy modification of the current experiment could be to change the extension of the molecule to study the buckling transition as the first plectoneme is formed. This would emphasize

the improved bandwidth of the optical tweezer detection and the ease of finding the buckling transition point using optical tweezers, two things that cannot be done using magnetic tweezers alone.

I did appreciate the modeling that went into analyzing the data in Fig. 5, but the comparison between theory and data is hard to evaluate. I'm not entirely clear how the experimental data has been processed.

CONCLUDING REMARKS

I do want to again emphasize that the instrument described has the potential to be useful for a variety of measurements. But the current manuscript leaves too many questions about reproducibility of the results for a reader to evaluate if the data is trustworthy. I therefore cannot recommend publication.

Reviewer 1:

Many thanks for your efforts in reading our report and offering very useful comments.

1. A comparison between the experiments and statistical mechanical calculations is made in the paragraph on pg 288. Do the authors have an idea for why there is a big peak in the denaturation probability in Fig 5b at ~7kpbs, which does not appear in the experiment. How well would these theoretical models be expected to perform in the context of these single molecule manipulation experiments?

In the context of these experiments, models such as Stress-Induced DNA Destabilization (SIDD) (see Benham & Bi *J. Comput. Biol. a J. Comput. Mol. cell Biol.* 2004) which we use here focus primarily on locating denaturation bubbles and are expected to be limited in its ability to accurately predict the locations of bubble formation for two reasons; 1) They assume that all torsional stress is partitioned into twist, effectively assuming that the DNA is fully extended, therefore over-predicting the prevalence of bubbles in systems in which plectonemes are present. 2) They fail to account for the possible influence of DNA curvature on plectoneme formation, and by extension, bubble formation. The presence of a peak in the region of ~7 kb is likely an artifact of this over-prediction of bubble formation, and although the peak is clearly present at ~7kb the integrated area under it is at least five times smaller than the integrated area under the suite of ~6 peaks that are pooled around ~4.5 kbp, so that probability for bubble occurrence at ~4.5 kbp will actually be over five times greater than at 7 kbp.

It is also worth noting that the presence of a plectoneme, and the implied presence of a co-localised bubble, at ~4.5 kbp does not exclude the possibility of a secondary bubble present at ~7 kbp. Further investigation studying in-depth the presence of single-stranded DNA would be required to directly locate denaturation bubbles. We have not added this important discussion in the main text (lines 313-326).

2. How well would these theoretical models be expected to perform in the context of these single molecule manipulation experiments?

With the caveat of the expected limitations of the SIDD algorithm as discussed in comment 1. above, the molecular dynamics simulation studies are likely to perform very well in the context of these single molecule experiments. We have used essentially the same simulation framework and parameterisation as for several previous recent single molecule DNA studies in which we see excellent agreement to AFM experimental imaging data (see Yoshua et al *NAR* 2021; Pyne et al *Nat Commun* 2021) concerning specific quantifiable topological features such as bending angles and radius of gyration, and note also optical tweezers and magnetic tweezers validation from several other single-molecule experimental labs (as discussed in the introduction), in addition to further validation against a range of bulk elastic properties of DNA demonstrated by the SerraNA algorithm we published previously (see Velasco-Berrelleza et al *PCCP* 2020). What our simulations struggle to predict well are the much longer time scale effects (e.g. above 10s of microseconds), since these of course require coarse-grained approaches. We now emphasised these important points in additional discussion in the manuscript (lines 470-477).

3. If possible, it would be helpful to replace the word "excellent" on line 292 with a quantitative measure of the performance of the calculations relative to the experimental data

We have replaced "excellent" in the text with a more quantifiable measure, specifically to indicate that the closest agreement between the experimental and predicted peaks is within 3.6% which we indicate on a modified Fig. 5b by the red triangles.

4. My understanding from the Online Methods section is that all of the atomistic MD simulations have been performed in implicit solvent. Such models can be extremely helpful for exploring conformational space rapidly in the absence of solvent friction, however, it is anecdotally believed that these models can destabilise dsDNA. Have the authors performed a control simulation to check that their simulations do not develop denaturation bubbles when the DNA is relaxed, and not subjected to torsional forces?

Yes, we did perform control simulations and confirm that we don't have bubbles in control simulations at any of the forces tried. We have now clarified this further in the revised text in online methods.

5. The implicit solvent models have been run for a very impressive (>2 microseconds) amount of time given the large (~300 bp) size of the DNA sequence. How many nanoseconds per day can be achieved on the computational resources used here, and where were the calculations performed? The restraints employed were clearly very technically challenging to implement, and I wonder how transferable these will be to other plectonemic DNAs.

Thank you. Yes, computational resources were performed on the local University of York Viking cluster using a single NVIDIA Tesla V100 GPU at a rate of 40ns/day. We agree with the Reviewer, with relatively standard local-level HPC resources other researchers should quite easily be able to implement our methods to other plectonemic DNAs. We have now added these details to online methods.

6. One comment on the thermodynamics: my understanding is that the formation of one bubble is favourable relative to forming two specifically because of the nucleation barrier, which in DNA is the same regardless of the length of the bubble (as discussed in papers by Benham).

That is our understanding also, which is alluded to in ref. 50 (Benham & Bi. *Comput. Biol. a J. Comput. Mol. cell Biol.* 2004) of our original submission. Considering the high energy cost associated with the initialisation of a run of strand separation of 10.84kcal/mol (see Benham, JMB, 1992), which is independent of bubble size, the formation of a single bubble is indeed favourable in this regime. We now discuss this important point in the revised manuscript, and have added this additional Benham reference (see lines 324-326).

7. I enjoyed the supplementary movies showing the behaviour of the DNA in the atomistic MD simulations. I found it helpful to play these at 50% speed, otherwise the movie was too fast to see the red denatured regions.

Thank you. We have followed your suggestion and now display these movies at 50% speed.

Reviewer 2:

Thank you for your efforts in appraising our work, we appreciate this and have aimed to respond positively to your questions, feedback and suggestions.

1. The manuscript lacks sufficient motivation to develop the technique from a biological standpoint. The authors should clarify what measurements can be performed by COMBI-Tweez that cannot be performed using existing methods.

Thank you for the opportunity to clarify the biological importance of the COMBI-Tweez technology, and in particular how it compares and contrasts with other complementary approaches. In essence, COMBI-Tweez enables the capability to perform highly controlled torsional manipulation on single filamentous biopolymers. We demonstrate this for both positive and negative supercoiling of single DNA molecules in real time, whilst transversely imaging the associated tether with fluorescence, but there are of course a range of examples of biomaterials with interesting torsional properties which could be investigated with this technology (e.g. see Emile et al *Nature* 2006 for an interesting review of silk protein) which we include expanded discussion of in the revised manuscript (lines 507-509). As the Reviewer indicates, there are of course other approaches available which deliver some, albeit not all, of these capabilities, at least in regards to experiments on DNA. In particular, i. an elegant method reported in King et al *PNAS* 2019 (ref. 25 of the original submission) which utilises the stochastic unbinding of biotin-avidin links between an optically trapped microbead and a DNA molecule in which the DNA is tethered between two such optically trapped microbeads, followed by a transient untwisting/relaxation in torsional stress, and subsequent biotin-avidin re-ligation event which thus enables negative supercoils to be introduced to the tether. This method can be used in conjunction with fluorescence imaging to visualize the DNA tether. It is valuable in not requiring the additional engineering and alignment of precise magnetic tweezers as with COMBI-Tweez, however, the addition of supercoils is fundamentally stochastic in nature and also limited to introducing only additional negative superhelical density as opposed to positive supercoils, and so there are specific aspects of COMBI-Tweez which do enable more control, fuller scope of both positive and negative supercoiling dependence investigation on topological effects of DNA, and higher precision reproducibility. ii. There is a method described in Van Loenhout et al *Science* 2012 (ref. 21 in our original submission) developed from the Cees Dekker lab which uses conventional permanent magnetic tweezers in a vertical geometry to twist a magnetic microbead conjugated to the free end of a surface-tethered single DNA molecule to induce putative plectoneme formation through introduction of positive supercoils, followed by rotation of the magnets to create an obliquely oriented tether which is almost but not entirely parallel to the focal plane of the microscope which can then enable subsequent observation of the dynamics of putative plectoneme structures via dye labelling of the DNA followed by epifluorescence microscopy. This is a neat approach as it does not require the high-precision engineering of the paired Helmholtz coils nor of the optical tweezers of COMBI-Tweez, however, there is no definitively high-precision clamping of force in this arrangement compared with COMBI-Tweez which controls force with high reliability and consistency using optical tweezers, there are potential issues associated with surface interference of the putative plectoneme structures, there is uncertainty as to the specific angle made between the extended tether and the coverslip surface due to limitations with the specific point of attachment on the magnetic bead and the associated height above the surface potentially leading to uncertainty concerning the directionality of mobility observations of putative plectoneme structures, and importantly this protocol does not enable real-time observation of

the formation of a plectoneme at the same time as the twisting of DNA is performed and so it is not possible to study specific features of early stage formation dynamics of plectonemes but is instead limited to exploring dynamics only once the plectonemes have been formed and only then several seconds after this. iii. There is also an elegant approach developed recently, also from the Cees Dekker lab (outlined in Ganji et al *Nano Lett* 2016, current ref. 22, but also further developed in an interesting Biorxiv preprint in Janiesson et al 2023 in which intercalation of Sytox Orange dye by surface-tethered DNA can induce torsional stress which is resolved in the DNA through plectoneme formation (which is associated with the introduction of additional positive supercoils into the DNA molecule) but also under certain concentration regimes of the dye to enable negative supercoils to be introduced into the tether following washing and associated removal of the intercalated dye. This is certainly a very valuable method since it is relatively easy to configure, allows for simultaneous visualization of the DNA tether and associated putative plectoneme structures, and can be performed with higher throughput than the COMBI-Tweez experiments. However, a key advantage of the COMBI-Tweez approach is that it can control the precise state of DNA supercoiling much more directly and consistently without relying on stochastic intercalating dye binding and unbinding events which, unavoidably, vary from tether to tether. Similarly, COMBI-Tweez can hold DNA tethers with controllable physiologically-relevant force which can be measured with high-precision, which is not the case the Sytox Orange intercalation method. Also, COMBI-Tweez enables observation of supercoiling directly which can be correlated to specific torsional manipulation of a single individual tether, which can be used, for example, to explore the dynamic formation of melting bubbles and associated plectonemes as opposed to being limited to, in-effect, steady state conformation measurements of DNA that are formed using the intercalation dye protocol. This is not to say that the intercalation method does not allow observation of putative dynamics of plectoneme once they are formed, however, it does not offer the capability to monitor the early stages of higher-order DNA structural motifs such as bubbles and plectonemes which COMBI-Tweez can. iv. Other valuable single-molecule manipulation methods, which are discussed and cited the introduction of the manuscript, which correlate either optical or magnetic tweezing with fluorescence imaging or have a very high precision of torsional control e.g. highlighted by a wonderful study from the Nynke Dekker team in Lipfert et al *Nat Meth* 2010, current ref. 17. However, none of these methods offer the technical capability of both constraining and manipulating the DNA supercoiling state in addition to independently controlling the tension and simultaneously permitting direct visualisation of the DNA tether. Although the correlative nature of the COMBI-Tweez technologically is more challenging to set up in terms of the engineering required compared to the methods highlighted in King et al 2019 and Ganji et al 2016 and, compared to Ganji et al 2016, is lower throughout, the technical capability of COMBI-Tweez to enable real-time observations of conformational and topological changes of DNA directly correlated to torsional manipulation at controlled force levels is a distinct and unique capability of this technology which enables substantive biological insights into the dynamic mechanisms of structural conformation changes to DNA. This in turn impacts understanding of multiple biological processes which affect DNA topology including DNA replication, repair and transcription. The methods highlighted in King et al 2019 and Ganji et al 2016 each offer important pros and cons compared to COMBI-Tweez, and we have now included substantial additional discussion of both of these elegant techniques to assist the reader in deciding for themselves which tools might offer best insights for different research applications – this has added more length to the manuscript, but we feel that this is required to openly and transparently inform the reader how these approaches can offer genuinely complementary insights (see lines 413-446).

One further point, but we have also now performed a range of additional experiments exploring the effects of elevated DNA concentration levels to controllably increase the DNA bead surface density coverage and intentionally induce greater than one tether to be formed between two microbeads; we already briefly alluded to the study of DNA braiding as one application of COMBI-Tweez in the original submission, but now we have been able to refine this to generate highly controllable braided lengths of two DNA molecules for in excess of 10 seconds of continuous high speed fluorescence illumination (see new Extended Data Fig. 2). This is a really exciting and unexpected feature of COMBI-Tweez because what these new data clearly show is that we can controllably generate braided DNA, and very precisely control the growing length of the braid interface by increase the number of bead rotations, and be able to directly visualise this growing interface in real time. This now opens new opportunities to explore DNA decatenation and un-bridging mechanisms, for example by using a range of additional topoisomerases in future experiments. Although there has been a recent preprint report which uses optical tweezers to probe DNA braiding (see <https://doi.org/10.1101/2023.11.20.567865>) there is no direct visualization of the growing braid in real time. This capability of COMBI-Tweez is a unique feature in itself (see new text lines 261-262).

Since the original submission we have explored the capability of COMBI-Tweez to torsionally manipulate and visualize much longer DNA constructs. We present examples of these new data in Fig. 5f-h which shows a typical torsionally constrained full length lambda DNA construct, over 3 times the contour length of our original 15kb test construct, indicating the presence of 3 plectoneme structures. Interestingly, after 800 ms from the start of fluorescence excitation there is a rapid increase of ~400nm in DNA extension indicative of a single-strand DNA nick, then followed by the sequential disappearance of the three plectonemes consistent with the propagation of a torsional relaxation wave from the nick site. We are not aware of any other reports from the alternative single-molecule torsional manipulation and visualization systems demonstrating this level of high-precision quantification of torsional waves propagating through DNA plectonemes. This capability opens up new opportunities in monitoring the rapid propagation of mechanical signals across single DNA molecules in real time.

2. This technique is interesting because it decouples the force and the torque hence enabling precise manipulation of the tethered molecule while sampling its response at kilohertz rates ... How is this decoupling important in the experiments demonstrated in the manuscript? What advantages does 50 kHz sampling rate provide in studying dynamics DNA plectonemes/ bubbles?

Thank you, this is an interesting question. We expect the frictional drag between a whole DNA tether of ca. 15kbp length and the surrounding aqueous solvent to be roughly comparable to that of a micron length scale bead, and so the typical whole molecule topological response time associated with changes to mechanical perturbation will be ca. milliseconds, so in that context the high frequency 50kHz sampling capability which is enabled by COMBI-Tweez's back focal plane QPD measurements are of limited value. However, in considering much shorter segments as opposed to the whole DNA tether, such as portions of DNA involved in small structural motif formation such as plectonemes/bubbles and in regions involved in DNA buckling transitions, or even just small length scale fluctuations corresponding to a few hundred bp in a tethered molecule, the effective drag is substantially lower by up to 2-3 orders of magnitude and so there is valuable information concerning topological fluctuations which can be extracted from the high-speed QPD data. We have now added exemplar analysis to illustrate this in the reviewed manuscript, in particular showing zoom-ins of the high-speed force signal calculated from the QPD data for

a tethered molecule undergoing DNA buckling (see zoom-in Fig 3b, lines 182-184) and for rapid structural dynamics over a ~100nm length scale in the plateau region of hat curves at high forces during undertwist (zoom-in Fig. 5a, lines 270-271). These data indicate measurable rapid fluctuations in the tether displacement which can be quantified with sub-millisecond sampling from the QPD signal demonstrating that high-speed features to conformational dynamics of DNA. In particular, with the zoom-in shown in Fig 2 the QPD data reveals that even though there is an overall trend in decreasing the tether end-to-end distance with continuous overtwist rotation of the magnetic bead, there is substantial heterogeneity between individual bead rotation cycles indicative of underlying high-speed fluctuations in DNA conformation, which are simply not detectable on the video rate measurements due to too low a sampling rate – this is a really interesting and useful feature of COMBI-Tweez, and many thanks to the Reviewer for suggesting we explore this further! Exploring how these high-speed dynamics are affected by different perturbations, e.g. ligand binding to DNA, is beyond the scope of this current manuscript which already includes substantial details of the characterization of this new technology, however, we do include additional important material relating to the application of high-speed monitoring of the disappearance of plectonemes in real time due to the propagation of a torsional relaxation wave following a nick in tethered full length lambda DNA (see lines 297-304).

3. The reviewer is of the opinion that the data presented in this manuscript can also be generated from experimental setup used in reference [21] at much higher throughput. The authors should provide a quantitative explanation justifying the uniqueness of a COMBI-Tweez measurement.

The Reviewer is referred to our response to their related comment 2. above. In essence, although we appreciate the elegance of the approach outlined in ref. 21 of the original submission, but it is incorrect to say that its technical capability is sufficient to enable all of the observations we report from COMBI-Tweez, and it is also incorrect to indicate that there is a substantive difference of throughput compared between either technique. In brief, Ref 21, Van Loenhout et al *Science* 2012, uses conventional permanent magnetic tweezers in a vertical geometry to twist a magnetic microbead conjugated to the free end of a surface-tethered single DNA molecule to generate plectonemes, followed by rotation of the magnets to create an obliquely oriented tether which is almost but not entirely parallel to the focal plane of the microscope which can then enable subsequent observation of the dynamics of putative plectoneme structures via dye labelling of the DNA followed by epifluorescence microscopy. There are clearly multiple and, in practice, non-trivial optimisation steps involved here in reality. See our response to your point 5. Below concerning throughput statistics for COMBI-Tweez, but in essence we calculate realistic throughput in COMBI-Tweez to be equivalent to 1 in 10 attempted tethers being a single tether and torsionally constrained to which, given the information available from ref. 21, we believe is not substantially different to that of ref. 21. and to suggest that this results in an improved throughput compared to COMBI-Tweez is simply incorrect, it should also be noted that due to the nature of all tethers being performed in ref. 21 only a single experiment can be done per slide, whereas using COMBI-Tweez a single slide can be reused innumerable times. There are issues also with this approach in having no high-precision clamping of force as there is in COMBI-Tweez, and unresolved issues associated with surface interference of plectoneme structures, uncertainty in the angle made between the extended tether and the coverslip surface due to limitations with the specific point of attachment on the magnetic bead and the associated height. Very importantly the protocol of ref. 21 does not enable real-time observation of the formation of plectoneme at the same time as the twisting of DNA is performed and so it is

not possible to study specific features of early stage formation dynamics of plectonemes but are rather limited to exploring dynamics only once the plectonemes have been formed and only then several seconds after this. COMBI-Tweez does have this quantifiable capability. We think the approach outlined. In ref. 21 has clear value in certain contexts, but it is not true that it can replicate all of the findings we outline from COMBI-Tweez. We have added important new discussion into the manuscript to compare and contrast COMBI-Tweez with several complementary single-molecule manipulation approaches (see lines 413-446) including those described in ref. 21, to enable to reader to decide for themselves which tool might best suit different specific research applications.

4. Line 55:

- Here, the references [23,24] cited by the authors refer to studies which DO NOT use a torsional constraint on the DNA, neither they use optically trapped magnetic beads not permanent magnets. The authors should revise this sentence correctly according to the cited papers or cite the correct reference that this sentence can be related to.

Thank you for noting this potential for confusion, we have now revised the text to make this explicitly clear (see lines 54-58).

5. In case OT with magnetic beads has been used for the first time in this study, the authors should mention it explicitly in the manuscript and provide control experiments showing that the magnetic bead does not absorb significantly in the NIR wavelength used in OT. Dissipation of heat from the irradiated magnetic beads can have a considerable effect on the DNA tether or its dynamics. This effect should be quantified and demonstrated to be insignificant over the time scales involved in typical MT or OT assays.

The Reviewer raises an important point – indeed, a limitation encountered by other research teams previously in attempting to use optically trapping of magnetic microbeads in biologically relevant contexts (personal communications with Nynke Dekker and team) has been local increases in temperature due to absorption of the near infrared (NIR) laser. We tackle this important question by first performing theoretical analysis to gauge what we would expect for the fall-off of temperature from a laser-heated bead into the surrounding solution. Theoretically on the grounds of perfectly spherical beads in an infinite medium assuming only simple thermal conduction, we find that this temperature decays decays with $\sim 1/x$ dependence where x the distance from the bead surface parallel to the microscope focal plane which is expected to hold for relatively small values of x less than the bead radius (see Supplementary Note 2). However, in reality, deviations from this are expected due to the axial and radial dependence of the power distribution, as well as due to a breakdown of an 'infinite medium' at the distance of the coverslip (which may act either as a heat sink or as a heat conductor in the plane); we therefore conducted finite element simulations to account for such effects (see Supplementary Note 3). To go beyond these theoretical estimates, we have experimentally measured heating effects by both attenuating the laser power and by determining the increase in temperature in the vicinity of optically trapped magnetic microbeads by determining the distance dependence on the melting transition for a range of purified alkane waxes with well-defined melting points, using a variation of a protocol originally developed by Scot Kuo (see Kuo, *Methods in Cell Biology* 1998) using non-magnetic microbeads. By first melting the waxes in buffer, sonicating and flash-freezing we create solid wax particles optimised to have effective diameters of typically a few to up to ~ 10 microns, which can be immobilised to the flow cell coverslip. By positioning an optically

trapped magnetic microbeads adjacent to a surface-immobilised wax particle we can then observe the wax melting transition in real time using brightfield microscopy with the liquid-solid interface clearly visible (see new figure, now Fig 2d-f). By repeating for several wax particles and trapped beads this allows precise quantification of the mean distance from the bead surface to the melting transition contour, which marks the wax's melting point, indicating temperatures of 43, 37 and 33°C at mean distances of approximately 0.3 μm , 2.7 μm and 4.1 μm respectively from the bead surface (see new Fig 2g). We use the experimental data corresponding to this small x condition (for the highest melting point wax we use, of melting point 43°C) using the $1/x$ interpolation to estimate that the bead surface temperature is 45°C. The finite element modelling of the heat transfer process outside the bead confirmed the deviation of the simple $1/x$ approximation for larger values of x , which we could account for by implementing a marginally higher surface temperature for the glass coverslip above room temperature, which is physically sensible in light of the relatively small micron distance between a trapped bead and the coverslip. This finite element model fits all wax data points within experimental error and indicates a range of approximately 45-30°C for distances from the bead surface from zero through to the anticipated $\sim 5 \mu\text{m}$ contour length of the DNA construct we use here, i.e. an average of $\sim 37\text{-}38^\circ\text{C}$ over this range. We don't claim to have designed it this way, however, it is clearly serendipitous that the increase in temperature is actually remarkably well aligned for performing physiologically relevant studies. We have included this important detail in the discussion of the revised manuscript, as well as highlighting the need to consider this increase in temperature when interpreting the dynamics of processes associated with DNA under study. We used transmission electron microscopy of 70 nm sections of embedded magnetic bead samples in conjunction with electron diffraction measurements (see new Fig. 2a-c) which confirmed the presence of a mixture of iron II and iron III oxides consistent with electron opaque magnetite in a thin shell of thickness $\sim 100 \text{ nm}$ around the surface of the beads. By incorporating this spatial patterning of magnetite into the finite element heat transfer model we could also compare this with other spatial distributions of magnetite throughout the bead (see Supplementary Note 3) which clearly indicates how a thin layer of magnetite results in substantively lower surface temperatures cover to locating magnetite closer to the bead core as occurs in several commercially available magnetic beads. The combination of having just a thin surface layer of magnetite with using a highly attenuated the laser power results in these relatively small, and potentially physiological useful, increases in sample temperature. We have now provided comprehensive quantitative characterisation data as the Reviewer suggests for the extent of this effect. This has now been added in full to the manuscript.

6. COMBI-Tweez being a new technique, should be benchmarked and compared with independently existing techniques. Some key quantitative details missing from the study are:
- What is the throughput of this technique? What is the typical number of successful tethers that can be analyzed? What percent of DNA tethers show single tethers or double tethers at given experimental conditions?

The Reviewer makes a valid point, and we note that it is this level of important detail which is sometimes absent from single-molecule manipulation studies. The typical throughput of the technique, as benchmarked by taking standard force-extension stretch-release data is equivalent to the formation of 1 tether trapped between two beads in 45.6% of attempts at forming a tether using our standard bead tapping protocol. It should be noted that this percentage is dependent on the concentration of DNA used, and in titrating between 0.06 and 1.2 $\text{ng}/\mu\text{L}$ we found this could be varied between forming tethers in every approximately 1 in 20 attempts to 2 in 3 attempts. We have now included additional analysis of these values by adapting theory reported in Leake at al *Biophys J* 2004 to indicate that these

results in a predicted fraction of binding events in which >1 tether is formed as low as 5% for the lower DNA concentration used. Measuring the percentage of DNA tethers showing single and two or more tethers given these experimental conditions was broadly consistent with this prediction respectively. When optimised, $\sim 20\%$ of single tethers are in principle useable for subsequent analysis. This level we deem a reasonable compromise to minimise the likelihood for multiple tether formation between two beads whilst still having an acceptable level of throughput – using an optimised $0.24 \text{ ng}/\mu\text{L}$ for DNA concentration which allowed to form typically one tether every 5 minutes if required. Interestingly, using higher DNA concentration level of $0.48 \text{ ng}/\mu\text{L}$ we are now able to consistent generated a small number (typically two) tethers between bead pairs to use to explore the effects of two DNA molecules twisting around each other visualized directly and in real time– this in itself is a unique innovation of COMBI-Tweez which no other existing technology has the capability to reliably explore. We have add this new detail into the revised manuscript (see lines 261-262, and online methods) Since submitting the original manuscript we have modified some aspects of the tether preparation protocol to improve the throughput (e.g. using only very gentle pipetting when working with the DNA construct following a valuable personal communication with Graeme King, UCL), as well as performing a wider titration of DNA concentrations to determine the optimum concentration level. We have written up these revised important details in full in the online methods.

7. Line 111: The authors do not provide any direct evidence of decoupling of force and torque. The use of magnetic beads in OT is not a common practice. Neither is it common to apply torsion to optically trapped magnetic beads. Control experiments should be performed, and their results added to Fig 1. or to SI. Two control experiments can be performed.
- i. To show that force (OT) is not affected by torque (MT) due to rotating magnetic field- monitoring XY position of the trapped bead while rotating the magnetic field by using torsionally unconstrained DNA as a tether.
 - ii. To show that torque (MT) is not affected by force (OT)- repeating “Hat-curve” experiments for various OT trap stiffness (varying intensities). These are my recommendations, however the authors can choose different control experiments to validate the decoupling.

Apologies for any lack of clarity here. Aspects of relevant control data concerning the decoupling of optical trapping force and magnetic field mediated torque were already published by us previously (see Zhou et al, *Photonics* 2015). We now include discussion of this, and for completeness we have now included additional extensive control data in the revised manuscript. To show that the optical-trapping force is not affected by the application of magnetic tweezers we monitor the XY QPD signal in the presence or absence of the rotating B field and using Fourier spectral analysis there is clearly no influence from the driving B-field signal on the OT force detection. Performing the experiment suggested by the Reviewer has limitations as suggested since even with no torsional constraint on the tether there is unavoidably bead precession due off inter-bead axis points of attachment of the tether on the rotating bead, which is obviously a geometrical effect of the tether-bead system as opposed to an influence of the MR on the OT. However, what we can do is perform Fourier spectral analysis to entirely remove this bead wobble component at the 1Hz driving frequency which shows no additional influence of bead rotation on the QPD signal (see Supplementary Fig. 5 and online methods). To clarify any misunderstanding here, however, we don't measure torque directly for DNA – we monitor the number of rotation cycles and can infer an indicative level of talk using a consensus torsional modulus estimate of $410 \text{ pN}\cdot\text{nm}$ for the DNA as we indicate in the original submission (see ref 2 of the original submission Bryant et al *Nature* 2003) but of course this value does not account for changes in torsional modulus due to the emergence of higher order structural motifs in the DNA upon

different twisting conformations, so should only be treated as indicative. However, we have used Hall probe measurements of the B-field, and added new fully quantitative calculations to estimate the realistic extent of any influence of the force applied to an optically trapped bead due to upper limit predictions for a non-zero B-field gradient in the vicinity of a bead which indicate a maximum likely contribution to the force on a magnetic bead from the magnetic tweezers of $<10^{-4}$ pN (see new Supplementary Note 1)

8. Line 182, 262: The authors observe negligible hysteresis in the rotation experiments. Does this mean that the magnetic bead can torsionally relax by thermal fluctuations? If so, this would lead to incorrect measurement of applied torque on the DNA molecule therefore and might be an important source of error that needs to be minimized.

No, we see no evidence for any such relaxation when rotating a bead in the absence of tethered DNA, we think it is unlikely that the magnetic bead can relax by thermal fluctuations. Also, and as Reviewer 3 observes (their comment 3.), there are examples in our hat curve data which are consistent with hysteresis, for example the hat curve data. There is a low level of unavoidable drift in the anchor bead position equivalent to ~ 15 nm/min (full details of which are now included in online methods) which does not affect short timescale measurements in bead rotation experiments over the timescale of ~ 1 min, but have a more significant effect when completing a full hat curve cycle equivalent to several tens of minutes. We now provide corrected hat curve data which aligns the start and end points of the hat curve cycles following such drift. There is also a small relaxation effect due to the time response of the force-clamp being limited to \sim a second. In addition, we sometimes observed bead pairs adhering transiently as very high level of DNA tether shortening due to induced DNA supercoiling which is manifest in an apparent relaxation effect in a portion of the hat curve shown in revised Fig. 3c (and see lines 218-220). In the hat curve protocols the magnetic bead is not torsionally relaxing – we know this because we can hold the bead in fixed orientation for extended periods and we don't see any indicators of relaxation over time in the absence of DNA.

9. Line 383: The authors claim to report first images of mechanically controlled bubbles: This claim is not completely valid because,

a. Mechanically induced plectonemes have been visualized before (reference [21], main text).

Ref 21 of the original manuscript does report visualisation of mechanically induced plectonemes, that is correct, however it does not report visualization of mechanically controlled bubbles, and also it does not report the formation of mechanically induced plectonemes directly correlated in real time to the mechanical driving torque. These are important nuances which we believe are important to clarify and so we have amended the text appropriately and fairly to avoid any perceived misconception in prior claims of originality in discovery (see lines 425-427).

b. The authors observe bubbles qualitatively as a localized blur. They should quantify these blurs and predict the number to base pairs inside the bubbles. To definitively claim that the blurs are because of ssDNA in the bubbles, a fluorescently labelled molecule that binds only to ssDNA (for eg. RPA- EGFP in ref. [23], main text) should be used to specifically image ssDNA bubbles.

We made significant efforts to get the experiment kindly suggested by the Reviewer to work both with fluorescently labelled RPA and Ssb, to the extent of having to exchange the surface chemistry of the anchor and magnetic bead and reoptimizing the coverslip passivation protocol to ensure adequately low levels of background fluorescent from the ssDNA fluorescent probes, with Ssb showing most promise initial from positive oligo controls and compatibility with the range of laser excitation and emission channels we have currently on COMBI-Tweez. The control experiments to a 100 nt ssDNA-cy5 oligo showed most promise in regards to cy3-SSB due to cy3 being smaller and more photostable than the EGFP that is fused to the RPA probe which we were able to try, and so we persisted with using the cy3-SSB on DNA tethers. However, the images on DNA tethers show no evidence for binding at comparable levels to the oligo positive controls. This in itself is an important finding – the minimum nucleotide base footprints that Ssb (or indeed RPA) has are approximately ~50nt (see Kim et al *Biochemistry* 1994; Bell et al *eLife* 2015). Our simulations under moderate conditions of negative supercoiling equivalent to a superficial density of 0.3-0.4, with low ~pN physiological level of force indicate a mean melting bubble length of ~60 nt, close to this footprint limit, where the probability of a bubble is predicted at around 10-20%. However, the method used to predict bubbles here has known limitations since it does not take into account the presence of plectonemes which should, in principle, lower the energy barrier to initial bubble formation due to the influence of the high localised curvature on the strength of the stacking interactions. This means that the energy barrier to bubble formation is lower at the tip of a plectoneme, thereby increasing the probability of a bubble forming and conceivably resulting in the formation of a much smaller tip-bubble. Our largely negative findings for the ssDNA probe binding, coupled with these simulation considerations may suggest that, at these very low forces levels which are 1-2 orders of magnitude lower than those indicated in the investigations by Gijs Wuite and team, the melting bubble size is not large enough to accommodate the typical ssDNA probe binding footprint. We have added these important discussions into the manuscript and a full description of the protocols tried and the negative results obtained in the revised manuscript (see lines 340-344 and online methods), and we hope that the Reviewer appreciates the significant lengths we have gone to here to act on their suggestion. However, we have also now fully quantified the localized blur which is indicative of melting bubbles through tether image segmentation and determination of the average line profile perpendicular to the tether axis, excluding any plectoneme foci. These analyses show that the line profile widths are ~20% larger, for $\sigma < 0$ compared to $\sigma \geq 0$. Although not directly proving that this is ssDNA, this result does add more quantitative support to this hypothesis, and we have included this in the revised manuscript (see revised Fig. 5d), but have been mindful to emphasize that this does not of course definitively prove that these regions are ssDNA.

Minor Comments:

10. There is very little reference to the relevant sections of online methods throughout the main text. The authors should refer to related sections of online methods and SI more frequently in the main manuscript.

We have now added in these references to online methods through the manuscript.

11. LINE 76: The authors claim that conventional MT introduces mechanical vibrations and limits the timescales that can be probed using MT. The authors should specify an upper bound for these timescales and compare it to the fastest time scales that have been probed in this study.

The basic design of conventional MT has not changed significantly since its original inception reported in Strick et al *Science* 1996 (ref 46 in the original submission), involving an inverted microscope with glass slide sample chamber and a pair of permanent magnets placed a few cm above this which can be mechanically rotated to induce a rotating B-field at the sample. This is the source of the mechanical vibration, with a typical maximum rotation frequency of the order of hertz (i.e. a timescale of ~seconds). The experiments we reported in the original submission used video rate sampling at 40ms per frame, and in our revised submission we show that the QPD back focal plane detection signal can be used directly to obtain sampling of force fluctuations at 50kHz (see insets in new Fig. 3b and Fig. 5a and associated text).

12. Line 150: What does lower OT stiffness refer to in the context of this study? The authors should specify the values of OT trap stiffness quantitatively and compare them with typical OT values used in similar experiments.

Apologies, we do indicate the OT stiffness in the online methods (nominal trap stiffness was set to approximately 10 pN/ μm) but for clarity we have now also added this to the man text.

13. Line 212: The manuscript presents stretching/twisting data from only 9 tethers. I find this number to be significantly lower to produce statistically significant data shown in Table 1. Is this number lower due to some experimental challenges? If yes, they should be mentioned.

The primary intention with these experiments was relatively basic quality control to demonstrate that the range of persistence lengths was broadly consistent with expectations from those reported by others for dsDNA previously, and that the range of contour lengths was broadly what we expect from the known construct sequence. There was no anomalous experimental challenge here as such. However, we take the Reviewer's point, and have added more data (we now show data from n=27 single tethers).

14. Line 246, "... a constant force above F_c to a SYBR Gold 247 labelled tether...". Here, what is the value of F_c ?

Apologies, the wording used in this sentence could have been improved. The value of F_c we take as being in the range 0.6-0.7 pN as per ref 45 from the original submission but we have reworded the section to indicate that the typical force level set is approximately 1-2pN, which is above this F_c level (see line 268-269).

15. Line 162 and Figure 2b. At point C, the bead exits the OT. In the figure 2b. one can see the tether length jump to $\sim 5.6 \mu\text{m}$ however in the inset, the bead is closer to the surface immobilized bead. Is the jump in the displacement to $5.6 \mu\text{m}$ at point C an artifact of the measurement? If so, it should be mentioned in the figure captions.

We usually observe a relatively small difference in height for an optically trapped bead (from 0.5-2 μm depending on the laser power and magnetite content) which is expected due to the axial NIR radiation pressure, which is thus manifest as a small change in height when a bead falls out of the trap. Note, we have modified this figure now to include additional QPD zoom-in data so only show inset from bead pairs which are still being actively trapped.

16 . Figure 2C.

a. There are no arrows indicating the path of the experiment in the figure like the authors have mentioned in the captions.

Thank you for pointing out this oversight, we have now added these (note, the new figure is now Fig. 3c, which also contains additional hat curve data).

b. There are two traces (four independent lines) in the plot. The authors should clarify what they mean or show them in different color with appropriate legend in the plot.

We have now clarified in the figure legend what these traces specifically mean and coloured the independent sections in different colours. Note also, the original graph showed the raw data uncorrected for drift effects, whereas we have now replaced these with drift corrected versions more appropriate in light of the extended duration timescale of several hundred second for these hat curve experiments (details described in online methods).

17. Line 234-235: How does observed 25% increase in contour length explain the 0.5 stoichiometry of the fluorescent dye? The authors should provide a brief justification for this observation or the reference [37] should be cited at the end of the sentence.

This value is consistent with the measurements reported previously in ref. 37 of the original submission, and we have now cited this reference at the end of the sentence (lines 256-257).

18. Figure 3 caption;

a. 3.c) Does $n=4$ refer to 4 stretch release cycle of the same tether? Or 1 cycle for 4 different tethers? In either case, is this number low because of the experimental constraints? In my opinion performing 30-50 stretch-release cycles instead of bootstrapping the data from the same cycle should be more meaningful.

Apologies for the misunderstanding; $n=4$ refers to 4 stretch-release cycles taken from the same tether. We agree with the Reviewer's suggestion and have instead used approximately 30 stretch release cycles performing a worm-like chain fit on each half cycle and generating the mean and s.d. of the persistence and contour lengths and reported this in the revised plot (note we did not detect any significant hysteresis between the stretch and release half cycles).

b. Even if the bootstrapping procedure gives negligible standard deviation, the authors should still include it in the manuscript. This is because in the jitter plots of contour length and the persistence length, they do observe a significant variance in the values calculated by the same procedure.

Agreed. The bootstrapping procedure as reported generates an SD of less than 0.01 nm, and we have now also included this in the figure legend.

c. The authors should point to the relevant sections in the online methods as they introduce new experimental or analytical technique.

Agreed, we have now implemented this.

19. Figure 4.a : arrows indicating the direction or a legend referring to the colors of the trace should be added to the figure. Such color scheme should also be implemented for figure 2.c

Agreed, we have now changed this (note, revised figures are now 5a and 3c)

20. Figure 4c and d: the panels could be independently labelled (for eg. with roman numerals) for more clarity and accurate referencing in the main text.

Agreed, we have now implemented this.

a. Figure 4.d, left most panel has a misplaced letter “c” in the image. This should be corrected.

Thank you for spotting, we have now corrected this.

b. Figure 4.d. The scale bar is different in different panels. I recommend making it uniform across all the panels.

Apologies for any misunderstanding here. The scale bar shown on the far left panel of what is now Fig. 5d is actually common to all of the panels. To make it more clear we have placed the scale bar above the panels and clarified in the figure legend.

21. Figure 5.d. Predictions of the simulation is that 40% of the positively supercoiled DNA show bubbles. This number is not 100% as in the case of negative supercoils but its significantly higher and hence demands an explanation.

We agree, this finding merits more explanation. There are two things worth mentioning here - first is that, while bubbles are observed in positive supercoiling, they are much more transient, never lasting longer than a couple of nanoseconds, whereas those in negative supercoiling are stable over the entire course of simulation. This difference in bubble timescale dynamics is most likely to cause of the difference in predictions of the simulations. But it is also worth noticing a second effect which may have some contribution which is that there is the possibility that the bubbles in positive supercoiling are the result of implicit solvent effects as base pairs tend to be easier to break in implicitly solvated DNA see online methods and lines 475-482).

22. SI: Figure 1. There appears to be a blue box in the top right corner which seems to be there by mistake.

Thank you for spotting this error, we have now corrected it (note, this is now SI Fig. 2)

23. SI: figure 6. Caption (Line 97) misspelled “instaneously” .

We have now corrected this error.

Reviewer 3:

1. Heating of the bead:

Typically optical tweezers will be used to trap polystyrene or glass beads that do not absorb much NIR light. The metal core of magnetic beads can absorb much more IR light, raising the possibility of heating the sample appreciably. The authors seem to have encountered this problem, since they mention that they “configured the OT stiffness to be relatively low compared to earlier studies of DNA that probed the over-stretch transition, allowing stable trapping over a range of a few pN relevant to the physiological scale while avoiding issues of localized heating of the trapped magnetic bead due to NIR laser absorption.” This heating issue has been a hurdle to other attempts to combine magnetic and optical tweezers, so the authors should specify how much heat is being created by the tweezers. Does this limit the maximum force that can be applied? Would it effect the kinetics of enzymes? Given the change in viscosity of water with temperature, measurements of the power spectrum should allow the authors to measure the amount of heating occurring near the sample.

The Reviewer raises some important points, similar to those of Reviewer 2 comment 4. We have now performed a fully comprehensive quantification of the temperature increase coupled to continuum and finite element analysis. The specific suggestion by the Reviewer has potential limitations: 1. measuring changes in the local viscosity will obviously only indicate a readout of the very local temperature increase near to the surface of the optically trapped as opposed to the full extent of the extended DNA tether, and 2. in addition to the heating effect, scattering of magnetite of the NIR laser potentially results in a reduction in effective optical trap stiffness which is non-trivial to then decouple from the temperature-dependent changes in local viscosity quantified from the corner frequency. Our alternative approach was to use the well-defined melting point of alkane based waxes. In essence (paraphrasing our response to Reviewer 2) we measure the increase in temperature in the vicinity of optically trapped magnetic beads by determining the distance dependence on the melting transition for three different purified alkane wax particles with diameters of up to 10 microns with well-defined melting points modifying protocol from Kuo, *Methods in Cell Biology* 1998. By positioning an optically trapped magnetic bead adjacent to a surface-immobilised wax particle we are able to observe the melting transition interface clearly using brightfield illumination (see new Fig. 2 and online methods). Across a population of beads and wax particles we could then quantify the mean distance from the bead surface to the melting transition contour, indicating temperatures of 43, 37 and 32°C at mean distances of approximately 0.3 μm , 2.7 μm and 4.1 μm respectively from the bead surface. Modelling the bead heat generation as a Laplacian in spherical polar coordinates indicates an analytical solution with $\sim 1/x$ dependence where x the distance from the bead surface parallel to the microscope focal plane which is expected to hold for small values of x less than a bead radius (see Supplementary Note 2). We use the wax data corresponding to the smallest x condition (for the highest melting point wax we use, of melting point 43°C) indicating that the bead surface temperature is 45°C. Finite element modelling of the heat transfer outside the bead confirms the deviation from the simple $1/x$ approximation for larger x , which we can account for by implementing a marginally higher surface temperature for the glass coverslip above room temperature, which is sensible in light of the relatively small micron distance between a trapped bead and the coverslip. This finite element model fits all wax data points within experimental error and indicates a range of roughly 45-30°C for distances from the bead surface from zero through to the $\sim 5 \mu\text{m}$ contour length of the DNA construct, an average of ~ 37 -38°C over this range. It is actually therefore remarkably well configured for performing physiologically relevant studies. We have included these details in full in the revised manuscript, as well as highlighting the need to consider this increase in temperature

when interpreting the dynamics of processes associated with DNA under study. We also used transmission electron microscopy of 70 nm sections of embedded magnetic bead samples with electron diffraction measurements (see new Fig. 2a-c and online methods) which confirmed the presence both iron II and iron III oxides consistent with electron opaque magnetite in surface shell of thickness ~100nm, which we incorporate directly into our finite element model and also compared with other hypothetical spatial distributions of magnetite throughout the bead (see Supplementary Note 3) which indicates how a thin layer of magnetite results in substantively lower surface temperatures closer to locating magnetite closer to the bead core as occurs in several commercially available magnetic beads. The max trapping force scales with laser power, so to achieve this acceptable lower level temperature increase limits our trapping force to approximately 10pN. This is well within the physiological range we want to explore for DNA, but we have clarified this in the revised manuscript (lines 158-160).

2. Reproducibility of OT:

The measured persistence lengths of the DNA shown in Fig. 2a vary by nearly an order of magnitude. In a typical optical tweezer instrument this would be considered far too high for reliable measurements. The authors make no attempt to explain this high variation. It could be caused by errors in the stiffness of the bead, drift in the position signal, or issues with the geometry of the setup, but some discussion of why this error is so high is needed to understand what other measurements would have the same high uncertainty. Also troubling is the fact that in Extended Data Fig. 1 there is a force extension curve where the DNA apparently gets shorter as the tension is increased. This may indicate drift is too high of a factor to even complete a force-extension curve.

The original submission indicating the distribution of persistence lengths was performed on a relatively small number of tethers as basic quality control to demonstrate broad agreement with a mean value of persistence length of approximately 50nm, which is consistent with findings from several other single molecule studies reported previously and cited in the manuscript. The Reviewer should note that these prior studies typically stretch DNA tethers using forces which are 1-2 orders of magnitude higher than the more physiologically-relevant range which we apply here, which does unavoidably result in greater variability in optimised worm-like chain fit parameters due to the associated smaller effective signal-to-noise ratio on the fitted data. However, in response to some of the other suggestions of Reviewer 2 we have modified our tether preparation protocol, performing a more comprehensive titration of DNA concentration values, optimising the DNA concentration to 24 ng/ μ L (which resulted in a probability of approximately 1 in 10 attempts to form a tether through the bead tapping protocol which ensured a relatively lower probability of ~5% for multiple tether formation which is of course a potential source for apparent variability in persistence length, as well as improving our bead mixing protocol using far more gentle pipetting following a valuable suggestion from Graeme King at UCL to minimise mechanical shear. Following these protocol refinements, we then generated more force-extension data, which has a comparable mean of ~50nm to that of the original submission but a lower variability. Note, drift is relatively low (see our response to Reviewer 2 comment 7 and also your comment 3. below) which we now report in the revised manuscript. There is variability in the trap stiffness likely due to variability in the thickness of the surface magnetite layer which can be seen from the TEM images and electron diffraction measurements which we now show (see new Fig. 2a-c) and this is why in general we measure the optical trap stiffness for each trap magnetic bead as opposed to relying on a mean value.

3. Apparent drift of magnetic tweezers:

The “hat curves” shown in Fig. 2c do not resemble typical hat curves on DNA. They exhibit far more hysteresis than has been reported previously, and as the superhelical density approaches zero they do not match up on the left- and right-hand sides. There seems to be drift in either the measured extension or the measured force that is biasing these results. Discussion is needed to characterize the source of these anomalies so readers can evaluate what other measurements could be affected. The authors also need to repeat the measurements on more than one DNA strand so the stability of the measured quantities can be evaluated.

The Reviewer is correct, the hat curve data was presented uncorrected for drift, which we have now rectified. There is a transient focal drift equivalent to a few μm on startup of the magnetic tweezers likely due to heating of the electrical coils which settles completely within a few tens of minutes, and so part of the experimental protocol is to ensure we have switched the coils on for at least 1 hour prior to any experimental measurement being taken. Beyond that, we measure only a low level of drift in XY and Z associated with the anchor bead of not beyond 15nm/min and 7nm/min respectively, and the optically trapped bead position itself has no directional drift but a maximum rms rate of displacement equivalent to 0.3nm/min. These relatively small drift levels cannot account for hysteresis in the short timescale bead rotation experiments obtained at each superhelical density datapoint. However, over the course of a full cycle of a hat curve of several tens of minutes this does account for most of the apparent hysteresis between the relaxed-to-twisted and corresponding reverse transitions of twisted-to-relaxed sections of the graph. In addition, there is also a small relaxation effect due to the finite response time of 1-2 seconds on the COMBI-Tweez force clamp, as well as transient sticking of beads at the start of the hat curve cycle which also causes the appearance of some hysteresis. We now show the drift corrected data which align the start and end points of each section of the hat curve (new Fig. 3c), and fully discuss the effects outlined above in associated text.

For completeness we include drift quantification details in the revised manuscript in full (see online methods). There is still some level of hysteresis remaining after correction, a source of which might relate to the emergence and subsequent relaxation of higher order DNA structural motifs. This would not be surprising since even before bubbles or plectonemes form we know that a range of sequence-dependent higher order transient motifs over a range of length scales do form upon twisting DNA, since we have observed these previously in molecular dynamics simulations specifically of twisted and extended sections of DNA (see Shepherd et al *NAR* 2020). In terms of comparison with hat curves from other studies, a very close inspection of the past reports of hat curves shows behaviour not dissimilar to our finding. For example ref. 44 of the original submission in Kiriegal et al *J Struct Biol* 2017, it is important to note that there are not a substantial number of hat curves reported (in ref. 44 there is just 1 shown for each of 3 different forces in the report's Fig 2) and actually looking closely at the torque plot of these data you can quite clearly see discontinuities, not dissimilar to those we observe, which we already show with more than one example. Similarly, in the original hat curve report of Strick et al *Biophys J* 1998 (ref 6 in our original submission) using a vertical magnetic tweezers approach, there is some hysteresis reported and discontinuities (see their Fig. 7) and not perfectly symmetrical hat curves (see their Fig. 9). One further point to note is that Fig. 2c left and right sections are after a full twisting up and untwisting cycle. There are also more marginal differences in bead rotation between this study and those reported previously which may influence hysteresis, and also differences in

bead-DNA conjugation chemistry. This latter point is important to note since our conjugation method has similarities to that employed in King et al PNAS 2019 (ref. 25 of the original submission) in using biotin conjugation, whose bond with avidin/streptavidin are known to stochastically unbind and religate as used to good effect in ref. 25 for introducing negatives supercoils – however, this could also be a source of discontinuity, and asymmetry/hysteresis. The larger step-like discontinuities e.g. between the different sections of the curves shown in new Fig. 3b, are consistent with not complete equilibration due to the relatively slow response time on the force clamp of ca. seconds. We did discuss this in the original submission, but to make it is an important point to note so we have clarified this further in the associated text. Note, we have now migrated additional hat curve data into revised Fig. 3c.

4. Strange discontinuities in magnetic tweezer data:

Both Fig. 2c and Extended Data Fig. 2 show a sharp corner as the buckling transition for plectoneme formation kicks in. This could indicate a potential use of the bandwidth of this instrument in study dynamics near this point, since the data seems to resolve the process in finer detail than a typical magnetic tweezers setup. However, Fig. 4a also shows sharp corner-like discontinuities at seemingly random locations. These bumps in the data make it hard to interpret what might be happening at the buckling transition.

Thank you, that's a good suggestion. This effect resonates with our response to your comment 3. above - our previous molecular dynamics simulations on extended dsDNA constructs (Shepherd et al *NAR* 2020) indicate the sequence-dependent emergence of higher-order structural motifs before the formation of bubbles or plectonemes due to twisting the DNA, and in excluding drift effects it is not unreasonable that there may be some relaxation of these emergent motifs between the discontinuity points on the hat curve. In looking at the high speed QPD force data we see that although there is a broadly linear trend to decrease the tether extension about overtwisting with each bead rotation, there is clearly significant heterogeneity from cycle to cycle e.g. some individual cycles fluctuate to higher values of extension instead of lower ones (see inset new Fig. 2b). In other words, this higher sampling does enable us to sample conformational heterogeneity which slow video-rate methods cannot do. As we note in our response to your comment 3. above, similar discontinuities have been observed by others previously. And as we also discuss above, the larger step-like discontinuities are a result of the comparatively slow response time of the force clamp. We have now added these details to the revised manuscript (see new Fig. 3 legend and online methods).

5. Final experiment fails to emphasize microscope's abilities:

The paper should include an example of an experiment that could only be done on the new instrument do demonstrate its usefulness. Unfortunately, the experiments described in Fig. 5 could also be performed on other instruments (e.g. the apparatus described in Van Loenhout et al. (ref. 21), which used magnetic tweezers to both twist and apply force to a fluorescently labeled DNA molecule). An easy modification of the current experiment could be to change the extension of the molecule to study the buckling transition as the first plectoneme is formed. This would emphasize the improved bandwidth of the optical tweezer detection and the ease of finding the buckling transition point using optical tweezers, two things that cannot be done using magnetic tweezers alone.

With due respect to the Reviewer, we disagree with their premise that all of the experiments described in fig 5 could also be performed on the apparatus described in Van Loenhout et al. (ref. 21 of the original submission). As we indicate in our response to Reviewer 2 comment 1, one of the unique features of COMBI-Tweez is its ability to monitor the emergence of higher-order structural motifs (such as bubbles and plectonemes, plus a range of other less stable emergent structures such as those identified in Shepherd et al *NAR* 2020) in real time, i.e. as the mechanical perturbation is actually imposed. It is simply not correct to suggest that the instrumentation of Van Loenhout et al. elegant as it is enables this capability since there is a delay equivalent to several seconds between the mechanical rotation imposed by the rotating magnetic field and then the subsequent method for rotating the permanent magnets to try to extend the DNA tether to be approximately parallel to the focal plane of the microscope, in which time several relaxation events could have occurred by will have been undetected. But we appreciate the spirit of the challenge set by the Reviewer. As we outline in our response to Reviewer 2 comment 8, we attempted to label the emerging ssDNA resulting from bubble formation, to further the unique technical capability, but this only worked very qualitatively due likely to the relatively small size of the melting bubbles at these very low forces being smaller than the required binding footprint. However, we have demonstrated that COMBI-Tweez has a valuable capability to sample forces in real time to a very high sampling frequency of 50Kz (see new Fig. 3b and 5a insets), which no other existing competing technology, especially not that outlined in Van Loenhout et al, is capable of. Also, since our original submission, we were able to perform more precise titrations of the DNA concentration used in incubating with beads, which has enabled us to reliably generate not only single tethers but also bead pairs with multiple tethers, which allows us to studying the twisting of two DNA molecules together and thus opens new opportunities to study catenation effect and ultimately the mechanisms of decatenation enzymes such as topoisomerases. Although other single-molecule approaches have reported braiding, e.g. in ref. 6 Strick et al 1998 of the original submission, there is no existing single-molecule precise force manipulation technology which enables this capability to create them and visualise *directly* in real time. We have now discussed these unique aspects of COMBI-Tweez in the revised manuscript since we agree with the Reviewer that it is important to allow the reader to clearly decide for themselves how the different technical capabilities of different force manipulation methods available complement each other and can be used to good effect for studying different biological processes. In addition, since our original submission we have expanded the capability of COMBI-Tweez to torsionally manipulate and visualize much longer DNA constructs, including full length lambda DNA, see new data in Fig. 5f-h. This example indicates the presence of 3 plectoneme structures. An intriguing observation is that after approximately 800 ms from the start of laser fluorescence excitation there is a rapid increase of ~400 nm in tether consistent with the formation of a single-strand DNA nick. This event is then followed by the sequential disappearance of the three plectonemes over the next ~200 ms consistent with a torsional relaxation wave diffusively propagating from the nick site. To our knowledge this is the first report of real time direct visualization single-molecule propagation a torsional wave in a single DNA molecule manipulation which again exemplifies a unique feature of COMBI-Tweez, opening new opportunities to investigate mechanical signal propagation in DNA in real time.

Reviewers' Comments:

Reviewer #1:

Remarks to the Author:

I am happy with the revisions that have been made to the manuscript so far. It is reassuring to know that the relaxed DNA simulations in implicit solvent do not denature. However, I was unable to find the data demonstrating this in the revised version of the manuscript, which I expected to see in Extended Data Figure 3, where the kymographs demonstrating denaturation of the supercoiled plectonemes are shown. Please could these data be included in a subsequent version of the paper, to convince the reader that the denaturation observed is indeed due to supercoiling, and not any artefact of the implicit solvent model.

Reviewer #3:

Remarks to the Author:

The updated manuscript has addressed my main concerns. I think the extra data the authors have integrated into the manuscript have helped assuage concerns about heating and successfully demonstrate the usefulness of the high bandwidth of this instrument.

Reviewer #4:

Remarks to the Author:

I am impressed by the efforts and the creative methods implemented by the authors to enhance the quality of this manuscript. The manuscript has been modified significantly to include key quantitative details and sufficient motivation for development of a new COMBI-Tweez technique. The authors first demonstrate the capabilities of the COMBI-Tweez technique and later analyze DNA supercoil dynamics using the correlative force-fluorescence-torsion approaches enabled by their experimental setup. The authors also support their findings with relevant MD simulations which enhance the understanding of the results.

However, this reviewer is of the opinion that in its current form, the manuscript appears to be a combination of two separate manuscripts: one which highlights the technique and other which highlights the dynamics of DNA plectonemes. The result of this combination in the current format is not satisfactory for the reviewer because in order to keep the length of the manuscript reasonable, it appears to miss out on highlighting the unique aspects of the technique as well as lacks sufficient quantification of the observations of DNA supercoil dynamics that can enrich the quality of the manuscripts if written separately. Therefore, in its current form, I do not recommend the publication of this manuscript in Nature Communications. This opinion is elaborated in the comments below.

Major Comments:

1. The introduction only focuses on motivation for developing COMBI-Tweez and does not include sufficient motivation for what biological problems the authors want to address by measuring the plectoneme size, position and the mobility of DNA plectonemes.
2. The authors have performed detailed calculations in context of the temperature profiles around an optically trapped magnetic bead (supplementary sections 2 and 3). These results are important stand-alone results, and it would be beneficial if they were summarized in the main text. However, in the current structure of the manuscript, they nest under the section "Stretching/twisting label-free DNA" which does not highlight the fact that this study demonstrates optical trapping of magnetic beads for the first time. Furthermore, the authors perform detailed theoretical analysis of the temperature profiles and also measure them experimentally. I suggest that the authors replace figure 1.h in the main text with supplementary figure 10.a as it demonstrates the agreement of the theoretical and experimental results. OR, overlay the expected theoretical temperature profiles in figure 1.g.

3. Figure 3.b, Lines 180-190:

What is the rate of twisting of the DNA in this experiment?

In figure 3.b, The author could overlay the force on top of the observed extension to better understand the bucking dynamics.

According to my knowledge, The non-linear buckling behavior with a fixed trap position is a new observation and can be investigated further in details. The authors observe that the DNA extension begins to reduce at ~ 50 seconds goes to a minimum ~ 60 seconds and then increases to the almost initial levels at ~ 70 seconds before exiting the trap.

- a. What are the statistics of these observations? How many tethers were analyzed Do all the tethers show same non-linear response and buckling at 2.2 pN?
- b. Because COMBI-Tweez is uniquely capable of performing this assay, it would greatly benefit the study if this response is examined in more detail for e.g. with respect to salt concentration
- c. Figure 3.b shows that the DNA tether length increases to 5.5 μ m however, in supplementary video 3, the trapped bead is seen to be pulled towards the anchor bead resulting in a smaller tether length after it escapes the trap. Can the authors correct for this discrepancy in the figure 3.b?

4. Conformational heterogeneity in rotation cycles.

The capability of observing rapid fluctuations (50 KHz) while applied torsional strain is a unique feature of COMBI-Tweez. This aspect has also not been analyzed in sufficient detail.

- a. How does the magnitude of these rapid fluctuations change with increasing supercoiling density in both directions?
- b. In Figure 3.b, the fluctuations appear to be larger in magnitude at higher forces (~ 60 seconds) as compared to that at lower forces? What can we infer from this?
- c. Is there a correlation between the plectoneme size or plectoneme diffusion coefficient and the magnitude of these fluctuations?
- d. How does salt concentration affect these rapid fluctuations?
- e. One could also perform PSD analysis on these rapid fluctuations with respect to changing superhelical density to gain more insights into the temporal dynamics of the plectoneme fluctuations.

5. Measurement of plectoneme diffusion dynamics.

- a. It is not clear why the authors decide to describe the dynamics of plectoneme fluctuations using a 2D diffusion model. The diffusion along the perpendicular component to the tether is because of the lateral fluctuations or 'wiggling' of the DNA as well as the plectonemes however, the parallel component corresponds to effective sliding of the plectoneme along the DNA. Both of these fluctuations are a result of different physical processes, and it is not clear if they can be combined into a single 2D diffusion constant.
- b. Figure 5.f,g,h: Are these experiments performed while continuously twisting the DNA or at a fixed torsional strain? The tethers must be held at a fixed supercoiling density to measure diffusion coefficients.
- c. For figure 5.h, lacks sufficient statistics to correlate the plectoneme size with diffusion coefficient. More than one tethers or plectonemes must be analyzed to draw conclusions that the diffusion coefficients are inversely proportional to the plectoneme size.
- d. What is the magnitude of force applied on the DNA during measurement of the parallel and perpendicular component of the plectoneme fluctuations? How do these components change with change in force?
- e. Figure 6.a: Do the authors observe the diffusion as well as hopping behavior in plectoneme dynamics as observed in Van Loenhout et al, Science,2012? Owing to the precise control in torsion and the force in COMBI-Tweez, the hopping behavior can potentially be examined more systematically.

Minor comments

1. Line 301 to 308: figure 5.f: What is the nature of supercoiling in this experiment? It is not very clear. I assume its positively supercoiled.
2. Figure5.f: What is the significance of the time in this movie? The authors can set 819 ms to 0

and then specify the times of other frames relative to the initial frames for more clarity.

3. Line 312: Does the number of base pairs in a plectoneme depend on the total tether length? Or the amount of supercoiling?

4. Table 1: Symbol for rmsd (perpendicular) is not printed correctly.

5. What is the biological significance of predicting the plectoneme position along a ~ 15 kb DNA tether? The authors should include the motivation for performing this measurement and simulations.

a. Figure 6.b.: How many tethers does the experimental data correspond to?

b. The fits from the Twist-DNA model appear to have a lot of 'spikes' which typically points towards overfitting that can also generate instabilities such as the peak at ~ 7.5 kb.

6. Line 434: The authors comment on the issues due to uncertainty in angle between the tether and the coverslip in permanent magnetic tweezers. The maximum angle in a 12.5 kb tether (~ 4.5 μm contour length) and a 1 μm magnetic bead considering the worst case of attachment point on the bead is $\arctan(1\mu\text{m}/4.5 \mu\text{m}) = \sim 11$ degrees.

In COMBI-Tweez, the authors use a 5 μm anchor bead and a 3 μm magnetic bead, In this case, for a 15 kb DNA (4.5 μm contour length), the typical angle between the tether and the surface can also be of the magnitude of ~ 10 degrees or higher simply because of the attachment point on the magnetic bead relative to the magnetization axis. Can the authors comment on this?

7. Line 302: Supplementary movie 11 probably refers to movie 12 in the latest version of the manuscript.

8. Line 451, Misspelled "quantify"

Reviewer #1:

I am happy with the revisions that have been made to the manuscript so far. It is reassuring to know that the relaxed DNA simulations in implicit solvent do not denature. However, I was unable to find the data demonstrating this in the revised version of the manuscript, which I expected to see in Extended Data Figure 3, where the kymographs demonstrating denaturation of the supercoiled plectonemes are shown. Please could these data be included in a subsequent version of the paper, to convince the reader that the denaturation observed is indeed due to supercoiling, and not any artefact of the implicit solvent model.

Many thanks again for reviewing our revised manuscript, it is enormously appreciated. We would like to express our regret for not including the relaxed DNA kymograph, which is now available in the Extended Data Figure 3. Furthermore, we neglected to note that, in order to eliminate brief occurrences, we included a temporal cutoff of 1 ns in our definition of the melting bubble. The method section now includes this. Three denaturation events are visible in the kymograph, but since they are shorter than 1 ns, they do not fit our description of melting bubbles. The caption of the Extended Data Figure now clarifies this.

Reviewer #3:

The updated manuscript has addressed my main concerns. I think the extra data the authors have integrated into the manuscript have helped assuage concerns about heating and successfully demonstrate the usefulness of the high bandwidth of this instrument.

Many thanks again, your time on this has been much appreciated.

Reviewer #4:

I am impressed by the efforts and the creative methods implemented by the authors to enhance the quality of this manuscript. The manuscript has been modified significantly to include key quantitative details and sufficient motivation for development of a new COMBI-Tweez technique. The authors first demonstrate the capabilities of the COMBI-Tweez technique and later analyze DNA supercoil dynamics using the correlative force-fluorescence-torsion approaches enabled by their experimental setup. The authors also support their findings with relevant MD simulations which enhance the understanding of the results.

Many thanks to the reviewer for taking a look at our revised submission – it is especially appreciated at such short notice and we have taken onboard your constructive points and made improvements where it is feasible to do so.

However, this reviewer is of the opinion that in its current form, the manuscript appears to be a combination of two separate manuscripts: one which highlights the technique and other which highlights the dynamics of DNA plectonemes. The result of this combination in the current format is not satisfactory for the reviewer because in order to keep the length of the manuscript reasonable, it appears to miss out on highlighting the unique aspects of the technique as well as lacks sufficient quantification of the observations of DNA supercoil dynamics that can enrich the quality of the manuscripts if written separately. Therefore, in its current form, I do not recommend the publication of this manuscript in Nature Communications. This opinion is elaborated in the comments below.

With due respect to this reviewer, we do not agree with their appraisal that our submission needs to be separated into two papers. However, what we will plan to do following, we hope, acceptance of our revised manuscript in Nature Communications, is to submit a subsequent Nature Protocols piece specifically on the back of the COMBI-Tweez submission, which can then specifically elaborate in more granular detail some of the points which have been carefully indicated by the reviewer, such as the temperature measurement methods they mention. We hope the reviewer will be supportive of this plan. We appreciate that this reviewer is raising a number of additional points beyond the original review process, some of which, with due respect to the reviewer, are simply not feasible to address at this stage – but we agree that they do make several valuable follow-up suggestions for subsequent studies, however, we are very clear that it is really imperative to publish the basic characterisation of the innovative instrumentation of COMBI-Tweez first. We have addressed the reviewer's points that are feasible to address at this stage to the best of our ability, and thank the reviewer for giving us this opportunity, which we believe has certainly improved the submission.

Major Comments:

1. The introduction only focuses on motivation for developing COMBI-Tweez and does not include sufficient motivation for what biological problems the authors want to address by measuring the plectoneme size, position and the mobility of DNA plectonemes.

As we state in the introduction, the use of DNA as test chiral biomolecule, and the associated measurement of plectoneme formation, is as a challenging test of the extensive technological features of COMBI-Tweez, as opposed to COMBI-Tweez having been specifically developed to uniquely address the sort of open biological questions which the reviewer highlights. However, we take the reviewer's point here, and have now elaborated more on the specifics of these open biological questions to ensure the reader is aware of these (see lines 86-87). In addition, to provide the necessary clarity to the reader that this manuscript primarily concerns the characterisation of COMBI-Tweez with the use of DNA as a challenging and appropriate test chiral biopolymer molecule, we have taken the decision to revise the manuscript title to more accurately reflect this.

2. The authors have performed detailed calculations in context of the temperature profiles around an optically trapped magnetic bead (supplementary sections 2 and 3). These results are important stand-alone results, and it would be beneficial if they were summarized in the main text. However, in the current structure of the manuscript, they nest under the section "Stretching/twisting label-free DNA" which does not highlight the fact that this study demonstrates optical trapping of magnetic beads for the first time. Furthermore, the authors perform detailed theoretical analysis of the temperature profiles and also measure them experimentally. I suggest that the authors replace figure 1.h in the main text with supplementary figure 10.a as it demonstrates the agreement of the theoretical and experimental results. OR, overlay the expected theoretical temperature profiles in figure 1.g.

We assume the reviewer refers to Fig. 2h and Fig 2g respectively, as opposed to Fig. 1h and Fig. 1g that they suggest in their comment above, since these Fig. 2 figures are the ones which indicate the appropriate temperature characterisation that the reviewer alludes to. We have now swapped Fig 2g and Supplementary Fig. 10a as suggested, and revised the text appropriately (lines 157-160). As we indicate in the paragraph starting on line 145 in which we allude to the issues encountered by others previously relating to temperature increases of optical trapping of magnetic beads, it would not be correct to suggest that we are the first

to use optical tweezers to trap magnetic beads. However, we take the reviewer's point here and thank them for the suggestion, so we have clarified in the introduction that this is the first report, to our knowledge, of stable optical trapping of magnetic beads that has been performed over physiologically relevant temperatures (lines 83-84).

3. Figure 3.b, Lines 180-190:

What is the rate of twisting of the DNA in this experiment?

This rate of twisting is already indicated clearly in the online methods - the bead rotation rate is 1 Hz throughout unless specified otherwise. But, to clarify for the reader, we repeat this in the main text at the point indicated by the reviewer (line 188).

In figure 3.b, The author could overlay the force on top of the observed extension to better understand the buckling dynamics.

We thank the reviewer for this interesting suggestion. However, this figure already includes substantive graphical content with the primary intention to demonstrate the enhanced technical capability of the high-speed back focal plane detection during the buckling transition. We tried a few iterations which included force detail in addition to the existing extensive details already shown and, unfortunately, we found with all of these that they significantly visually detract from this primary message. So, we have decided to retain the figure as is, but thank the reviewer for the suggestion.

According to my knowledge, The non-linear buckling behavior with a fixed trap position is a new observation and can be investigated further in details. The authors observe that the DNA extension begins to reduce at ~ 50 seconds goes to a minimum ~ 60 seconds and then increases to the almost initial levels at ~ 70 seconds before exiting the trap.

a. What are the statistics of these observations? How many tethers were analyzed Do all the tethers show same non-linear response and buckling at 2.2 pN?

We agree with the reviewer, this is an interesting observation. We obtained data comparable to that shown in Fig. 3b for five different tethers in which we were able to overtwist to such an extent that the magnetic beads were ultimately pulled from the optical trap, all at the same high force level of ca. 1-2 pN. Yes, these all show a broadly similar shaped non-linear buckling response in reaching a local minimum extension upon overtwist (we have clarified this now in the figure legend) – for interest, we show one of these additional traces in Rebuttal Fig. 1 below:

Rebuttal Fig. 1 Additional example of non-linear buckling response upon prolonged overtwisting of DNA (red data shows median filtered data of the QPD response). The magnetic bead is pulled from the optical trap in this example at >120 s.

We have now added in additional text in the manuscript to clarify this intriguing buckling behaviour (lines 188-194).

b. Because COMBI-Tweez is uniquely capable of performing this assay, it would greatly benefit the study if this response is examined in more detail for e.g. with respect to salt concentration.

We agree with the reviewer that exploring the granular details of DNA buckling transitions, for example including wide ranges of ionic strength characterisation, would be potentially valuable as a follow-up to our innovative technological development work here. However, this is simply not something we can address unfortunately in a concise and timely manner in this current manuscript. We did perform some qualitative optimisation using a range of different buffering conditions in the early stages of the project, which spanned a range of ionic strength from ~10 mM up to ~200 mM, so we can at least confirm that the technology has the capability to tether over a range of different ionic strength conditions – we have now added text to clarify this for the reader (lines 139-142). However, to perform the type of methodical study of ionic strength the reviewer is suggesting would really require new engineering of multi-channel microfluidics, and this we feel would be an unreasonable and unfair delay to dissemination of our innovative technological developments here. We impress on the reviewer that the primary intention of our seminal manuscript here is to articulate and characterise the innovation of the COMBI-Tweez technology exemplified through a challenging and appropriate example of a chiral biomolecule of DNA. This we believe is clear

from the abstract summary and the introduction. However, as we indicate in our response to the reviewer's point 1. above, to make this intention absolutely explicit to the reader we have now revised the title of the manuscript to better reflect this primary intention. However, we believe the reviewer makes valid and useful suggestions for future studies, and so we have added new mention in the discussion that applying this new technology to delineating the granular, mechanistic details the reviewer alludes to could be a valuable future route to explore (lines 544-556).

c. Figure 3.b shows that the DNA tether length increases to 5.5 μ m however, in supplementary video 3, the trapped bead is seen to be pulled towards the anchor bead resulting in a smaller tether length after it escapes the trap. Can the authors correct for this discrepancy in the figure 3.b?

The trapped bead in question is being pulled towards the optical trap after the buckling transition, which is at times >75s on the Fig. 3b plot - the magnetic bead has been pulled out of the optical trap, and so there is no correction which can be applied due to the lack of any sensible measurement after this point. But, to make this clearer to the reader, we have now added clarification of this to the main text (line 191-192).

4. Conformational heterogeneity in rotation cycles.

The capability of observing rapid fluctuations (50 KHz) while applied torsional strain is a unique feature of COMBI-Tweez. This aspect has also not been analyzed in sufficient detail.

a. How does the magnitude of these rapid fluctuations change with increasing supercoiling density in both directions?

That's an intriguing question. Increasing the superhelical density from $\sigma = 0.02$ to 0.04 increases the amplitude of the rapid extension fluctuations by a factor of approximately two (this can be seen on the inset of Fig. 3b on the filtered red trace). With undertwisting at comparable forces we don't observe any buckling transition but simply a broadly constant tether length during underwist which is consistent with DNA melting as observed by others (e.g. the hat curves in Meng et al *Biophys J* 2014, et al *Science* 1996, references 48 and 49 respectively in our manuscript). We have now included additional text to clarify this (lines 188-190).

b. In Figure 3.b, the fluctuations appear to be larger in magnitude at higher forces (~60 seconds) as compared to that at lower forces? What can we infer from this?

Yes, we agree with the reviewer, as we indicate in our response to point 4a above, there is an increase in the amplitude of the rapid extension fluctuations by a factor of roughly two between the lower force regime at $\sigma = 0.02$ to the higher force regime at $\sigma = 0.04$. It is very hard to determine the source of this behaviour without performing extensive additional investigations which we believe are beyond the scope of this current manuscript. However, since the reviewer has invited us to speculate, one possibility is that a source of these ~100 nm fluctuations could be rapid transitioning between relatively small sections of DNA which form metastable structural states. Although performed in different absolute regimes of force and twist, we observed qualitatively similar types of features previously in all-atom molecular dynamics simulations which emerge during twisting and stretching of DNA (see Shepherd et al, *NAR* 2020, reference 31 in our current manuscript). Such putative transitions might conceivably act between compact structural motifs and the "normal" non-compact extended states. Therefore, during overtwist the force experienced the ends of these motifs will increase, and so increasing this force conceivably increases the fluctuation amplitude by increasing the extension of the non-compact section of DNA in acting as an entropic spring in the usual manner. However, to really substantiate this hypothesis would require very

significant further methodical study (e.g. examining the sequence dependence of difference DNA constructs in revealing the emergence of differently structured metastable states), which we feel is beyond the scope of our current manuscript. However, we have included additional speculation as to what the source of this amplitude dependence might be along the lines of what we indicate above (lines 197-199). And, we have also indicated in the discussion that this would be an interesting route for future study (lines 548-551).

c. Is there a correlation between the plectoneme size or plectoneme diffusion coefficient and the magnitude of these fluctuations?

That's an interesting question, but unfortunately we cannot address this question directly with our current approach. The magnitude of the fluctuations is correlated with changes in force applied to the tether, as discussed above; however, in the plectoneme-forming experiments, we require force clamping, and so it is not possible to investigate how fluctuations change with the applied force on the same tether while generating a plectoneme. We can say that there may be future scope for modifications of the technology to enable this in the future, however, and we indicate this in the discussion (line 543).

d. How does salt concentration affect these rapid fluctuations?

As we indicate in our response to the reviewer's comment 3b above, although we performed qualitative tethering during the optimisation stages of the project using different buffering conditions, to properly address the reviewer's question here would require extensive new engineering of the instrumentation which we feel is beyond the scope of our current manuscript and would delay dissemination of our technological developments unfairly.

e. One could also perform PSD analysis on these rapid fluctuations with respect to changing superhelical density to gain more insights into the temporal dynamics of the plectoneme fluctuations.

Thanks to the reviewer for the suggestion, that's helpful. We have now performed power spectral analysis of the buckling transition data, and for information include the additional plots used in the Rebuttal Fig. 2 below:

Rebuttal Fig. 2. a. Mean power spectra for data of main text Fig. 3b. A sliding window of 10 s width was moved in 0.6 s steps from $t = 0$ s to $t = 75$ s to obtain 108 QPD signal segments. A Fast Fourier Transform was then performed on each followed by taking the absolute value and squaring to obtain 108 individual power spectrum density distributions (PSDs). We then randomly selected 25 PSDs from $t = 0$ s to $t = 45$ s and obtained the arithmetic mean average to represent the mean PSD for the pre-buckling region (red).

Another 25 PSDs were selected from $t = 50$ s to $t = 75$ s and mean averaged to represent the buckling part (blue). **b.** Difference plot of the mean buckling and pre-buckling power spectra indicated in panel a. The maximum difference is in the low frequency region calculated between 2 Hz (to avoid any contamination from the 1 Hz bead rotation driving frequency) and approximately 60 Hz, resulting in an increase in the time-integrated PSD of approximately 85% for the buckling compared to the pre-buckling regions.

In conclusion, although the buckling and pre-buckling power spectra look broadly similar, there are some subtle differences, most evident in the lower frequency components <60 Hz but above the level of the 1 Hz driving frequency, manifest as increase in the time-integrated power spectral density of approximately 85% for the buckling compared to the pre-buckling mean power spectra. To determine the definitive source of this difference would require far more extensive study which we believe is beyond the scope of this current manuscript, however, again since the reviewer is inviting us to speculate, we can suggest that the cumulative emergence of supercoiled DNA structures during buckling caused by prolonged overtwist could conceivably result in a higher overall frictional drag of the bead-DNA systems which might therefore result in an increase in the amplitude of the lower frequency components (put another way, since the effective corner frequency is lower following prolonged overtwist then the effective frictional drag of the system must be higher since the optical trap stiffness is constant throughout). We have therefore added this speculation to the manuscript (lines 201-204). But, as the difference plot of Rebuttal Fig. 2b suggests, there are also smaller but significant changes in higher frequency regions, and so clearly there could be multiple effects resulting in these differences which would very much require substantive further future study to properly understand, which we have now indicated in the discussion (lines 552-553).

5. Measurement of plectoneme diffusion dynamics.

a. It is not clear why the authors decide to describe the dynamics of plectoneme fluctuations using a 2D diffusion model. The diffusion along the perpendicular component to the tether is because of the lateral fluctuations or 'wiggling' of the DNA as well as the plectonemes however, the parallel component corresponds to effective sliding of the plectoneme along the DNA. Both of these fluctuations are a result of different physical processes, and it is not clear if they can be combined into a single 2D diffusion constant.

Converting the mean square displacement measurements to an effective 2D diffusion coefficient is simply the least unbiased way we have to quantify apparent local plectoneme diffusion in the 2D plane in which we visualise the tether. But, we agree that the reviewer highlights a useful distinction. As the reviewer correctly articulates, there may well be different biophysical processes responsible for the mobility parallel and perpendicular to the tether axis. However, it is not our intention here to explicitly explore the sources of either of these different biophysical processes. Therefore, to clarify to the reader, we have now revised the wording concerning the 2D diffusion coefficient estimates to explicitly reflect this important detail that the parallel and perpendicular components are likely to be due to different physical processes (lines 334-338).

b. Figure 5.f,g,h: Are these experiments performed while continuously twisting the DNA or at a fixed torsional strain? The tethers must be held at a fixed supercoiling density to measure diffusion coefficients.

Yes –diffusion coefficients are measured at fixed supercoiled density values. As we indicate at the beginning of the paragraph which describes the experiments highlighted in Fig. 5 these experiments are performed using a discontinuous imaging protocol in which the tether

is typically visualised stroboscopically for 10 datapoints at distinct values of superhelical density. Bead rotation occurs between these stroboscopic illumination timepoints.

c. For figure 5.h, lacks sufficient statistics to correlate the plectoneme size with diffusion coefficient. More than one tethers or plectonemes must be analyzed to draw conclusions that the diffusion coefficients are inversely proportional to the plectoneme size.

With due respect to the reviewer, we are very careful to indicate “the example of” and “In this example” (see lines 299, 320 and 323). To clarify, at no point are we making the claim in this figure that the behaviour of a lower plectoneme diffusion coefficient is associated with a larger plectoneme size across a population of many molecules and many plectonemes. However, to explicitly make this very clear to the reader, we again add the phrase “in this example” to the associated figure legend of Fig. 5g (please note, we have swapped Fig. 5g and Fig. 5h in this revised version of the manuscript to be consistent with their order of citation in the main text).

d. What is the magnitude of force applied on the DNA during measurement of the parallel and perpendicular component of the plectoneme fluctuations? How do these components change with change in force?

As we indicate in online methods, the orientation of DNA tethers is adjusted to be parallel to the optical trapping force, while the perpendicular component is set to a zero mean level which, of course, fluctuates around the Langevin force several orders of magnitude lower than the parallel component. For the 15kbp construct, low force tethers of Fig. 5 were set at approximately 0.5 pN, while the high force tethers were set at approximately 1 pN. For the much longer full-length lambda DNA construct, the tether shown in Fig. 5f is extended to a fractional extension of approximately 70% (equivalent to a low force of approximately 0.3 pN). We have now indicated these values in the appropriate figure legend.

e. Figure 6.a: Do the authors observe the diffusion as well as hopping behavior in plectoneme dynamics as observed in Van Loenhout et al, Science,2012? Owing to the precise control in torsion and the force in COMBI-Tweez, the hopping behavior can potentially be examined more systematically.

That’s a really interesting question. As the reviewer suggests, Van Loenhout et al Science 2012 (reference 22 in our revised manuscript) reports an intriguing plectoneme hopping process that appears to facilitate rapid long-range plectoneme displacement by nucleating a new plectoneme at a relatively distant position in less than ~20ms. We don’t observe any direct evidence for this in our current work, however, as we have clarified to the reviewer in our previous responses, our primary intention with this current manuscript is to present and characterise the COMBI-Tweez technical capabilities as opposed to providing specific systematic studies of focused biological questions relating to plectonemes. But, as the reviewer suggests, this could be another potential valuable direction for a future application of COMBI-Tweez, and we now add mention of this to the discussion (lines 553-554). However, what we can say qualitatively in relation to the reviewer’s question is that there is some suggestion of anticorrelated behaviour in portions of the intensity vs time traces for the plectonemes measured in Fig. 5h - in particular, the blue and cyan traces which show some qualitative suggestion of anticorrelation in places. The phase shift of this qualitative anticorrelation is equivalent to a few tens of ms over a lengthscale of a few microns. Although this is not direct evidence for plectoneme hopping behaviour per se this may at least be qualitatively consistent with long lengthscale interaction between plectonemes over roughly the same timescale as reported by Van Loenhout et al 2012. We have added a few words of description of this to the figure legend, but we are cautious against

overemphasizing and over interpreting this observation since it is clearly qualitative.

Minor comments

1. Line 301 to 308: figure 5.f: What is the nature of supercoiling in this experiment? It is not very clear. I assume its positively supercoiled.

The tether was imaged following a series of positive bead rotations, yes. We have now indicated this in the revised manuscript (line 320).

2. Figure 5.f: What is the significance of the time in this movie? The authors can set 819 ms to 0 and then specify the times of other frames relative to the initial frames for more clarity.

0 ms is set to be the start of the fluorescence acquisition (i.e. the start of the associated Supplementary Movie 11), while 819 ms (for the first image panel shown on the left of Fig. 5f) is simply the time relative to this corresponding to an image frame which clearly shows well the three plectonemes a very short time just before the single strand nick event. We have now clarified this in the revised text of the associated figure legend.

3. Line 312: Does the number of base pairs in a plectoneme depend on the total tether length? Or the amount of supercoiling?

The 10% value indicated in this line refers to the number of bp in the total tether length – we have now revised this wording to make this clear (line 331 of revised manuscript).

4. Table 1: Symbol for rmsd (perpendicular) is not printed correctly.

We think this must be a pdf conversion error from the manuscript submission portal since it appears OK our end. However, we have re-inserted the perpendicular symbol in question and double checked, but we will also make sure to double check this at the proofing stage as appropriate if this is still an issue or not.

5. What is the biological significance of predicting the plectoneme position along a ~ 15kb DNA tether? The authors should include the motivation for performing this measurement and simulations.

The significance of the plectoneme position calculations using a 15kbp construct is that this is the same sequence as the 15kbp used in the experiments, and so presents a very good explanation of our experimental observation fluorescent puncta at positions which agree well with those predicted in the simulations (as indicated very clearly in the earlier comments of reviewer 1). We have now clarified this in the text (line 344-345).

a. Figure 6.b.: How many tethers does the experimental data correspond to?

The blue trace of Fig. 6b corresponds to a single tether measurement. We have now clarified this in the revised figure legend.

b. The fits from the Twist-DNA model appear to have a lot of 'spikes' which typically points towards overfitting that can also generate instabilities such as the peak at ~7.5 kb.

No, these aren't overfitting spikes – there is no "fitting" performed here per se. SIDD model uses a series of parameters related with sequence-dependence melting and bubble initiation, which are constituent to the model and were obtained by classic standard experiments (SantaLucia et al Biochemistry 1996 35:3555-62), which are totally independent from the ones presented here. These constants were not altered in our calculations. We

realized that there was a sentence in the discussion section that might give this wrong impression, which has been corrected in the new version. For clarifying this misinterpretation, we have changed the term “simulations” for “calculations”, when referring to the use of SIDD model (lines 510-511). The appearance of spikes in the black trace of Fig. 6b is a consequence of the way in which the energetics work. There is a fixed nucleation energy (~12kcal/mol) associated with forming a bubble that is entirely independent of the sequence in question or the bubble size. As such it is much more energetically favourable to form a small number of large bubbles, rather than a large number of small ones. This means that, when a sequence is identified as being favourable for bubble formation, it becomes incredibly likely that a bubble will form there, hence a very high probability (i.e. a “spike” in the trace). In our case, the supercoiling density is high enough that there are several of these regions. This really is a fundamental feature of the SIDD model - it identifies a relatively small number of regions in which bubble formation is highly likely (see the figure 1 here for example <https://academic.oup.com/bioinformatics/article/31/3/421/2365978>).

6. Line 434: The authors comment on the issues due to uncertainty in angle between the tether and the coverslip in permanent magnetic tweezers. The maximum angle in a 12.5 kb tether (~ 4.5 um contour length) and a 1 um magnetic bead considering the worst case of attachment point on the bead is $\arctan(1\text{um}/4.5\text{ um}) = \sim 11$ degrees.

In COMBI-Tweez, the authors use a 5 um anchor bead and a 3 um magnetic bead, In this case, for a 15 kb DNA (4.5 um contour length), the typical angle between the tether and the surface can also be of the magnitude of ~ 10 degrees or higher simply because of the attachment point on the magnetic bead relative to the magnetization axis. Can the authors comment on this?

In the case of using the “permanent magnet assay” (reference 22 of our revised manuscript), one end of the tether is bound to the surface of a coverslip while the other is bound to a magnetic bead. The primary source of the uncertainty is not having direct and definitive information as to where the position of the coverslip surface is relative to the bead since in practice the way these experiments are done is by trying to touch the bead onto the coverslip surface to determine roughly where the surface is, but this is inevitably going to be imprecise as one must rely on essentially by-eye observations of relatively small defocusing movements of the bead in z. With COMBI-Tweez, conversely, we focus on the midpoint of both beads which is relatively easy and precise for the relatively large beads we use, so do not have this type of problem. We have clarified this in the revised text (lines 457-458).

7. Line 302: Supplementary movie 11 probably refers to movie 12 in the latest version of the manuscript.

Well spotted, thank you to the reviewer! We have now corrected this oversight (lines 309-311 and line 320 in revised manuscript).

8. Line 451, Misspelled “quantify”

Corrected, thank you to the reviewer (line 480, revised manuscript).